# Metabarcoding Analysis of Rhizosphere and Bulk Soils in Bulgaria Reveals Fungal Community Shifts Under Oat–Vetch Intercropping Versus Sole Oat Cultivation

**DOI:** 10.3390/microorganisms14010042

**Published:** 2025-12-24

**Authors:** Stefan Shilev, Mariana Petkova, Ivelina Neykova

**Affiliations:** Department of Microbiology and Environmental Biotechnologies, Agricultural University–Plovdiv, 12 Mendeleev Str., 4000 Plovdiv, Bulgaria; marandonova@gmail.com (M.P.); ivababrikova@abv.bg (I.N.)

**Keywords:** rhizosphere, mycobiome, intercropping, green manuring, soil health, high-throughput sequencing, fungal community structure

## Abstract

Fungal communities in the rhizosphere are crucial in maintaining soil health, driving nutrient cycling, and enhancing plant productivity. This study examined the role of intercropping of oats (*Avena sativa* L.) with vetch (*Vicia sativa* L.) and their subsequent use as green manure (incorporating fresh plant biomass into soil to enhance nutrient cycling and microbial activity) on fungal diversity and community structure. Three field treatments were organized as follows: (i) unplanted control, (ii) single-oat cultivation, and (iii) oat–vetch intercropping. In the ripening stage of oats development, the plants in the intercropping treatment were ploughed at a depth of 30 cm as green manure. Soil samples at ripening stage and 3 months after ploughing were analyzed. High-throughput sequencing of the ITS2 region, combined with multivariate diversity analyses (alpha and beta diversity, PCA, NMDS, and UniFrac), revealed distinct fungal community profiles across treatments. Ascomycota dominated under conventional and untreated conditions, while Basidiomycota, Mortierellomycota, and Glomeromycota were enriched in intercropped and organically amended plots, notably at intercropping. Intercropping and green manuring significantly increased species richness, evenness, and phylogenetic fungal diversity. These treatments also supported higher abundances of beneficial fungi such as *Mortierella*, *Glomus*, and *Trichoderma*, while reducing potentially pathogenic taxa like *Fusarium*. Rank–abundance curves and rarefaction analysis confirmed that diversified systems hosted more balanced and complex fungal assemblages. Beta diversity metrics and ordination analyses indicated strong dissimilarities between the conventionally managed and diversified systems. The results showed that intercropping and organic inputs alter fungal community composition and promote microbial resilience and ecological functionality in the rhizosphere. These practices promoted the development of stable and diverse fungal networks essential for sustainable soil management and crop production.

## 1. Introduction

Modern agriculture has historically favored monoculture systems to maximize yield, but such practices often degrade soil quality, reduce microbial diversity, and increase the incidence of soil-borne diseases [1]. Monocultures limit root diversity and exudation patterns, which disrupt vital ecological processes, impair long-term soil fertility, and reduce ecosystem resilience. In contrast, diversified cropping systems, such as intercropping, cover cropping, and the incorporation of green manure, offer sustainable alternatives by enhancing plant diversity, nutrient cycling, and soil biological activity [2,3]. Recent field studies also provide new evidence on oat–vetch systems and soil microbiomes. Oat–vetch mixed silage increased microbial diversity and metabolite production in alpine pastures [4], while a field trial in Mongolia demonstrated that different seeding ratios of this intercropping practice influenced biomass, soil nitrogen, and potentially fungal community structure [5]. These recent findings strengthen the hypothesis that oat–vetch intercropping promotes a more balanced soil microbiome and positive plant–microbe interactions under field conditions.

Legume-based intercropping introduces plant species with complementary root structures and exudate profiles, shaping the rhizosphere environment and influencing microbial recruitment [6,7]. These plant species interactions create niches that support beneficial microorganisms, improve nutrient dynamics, and foster a more balanced soil microbiome [8]. Green manuring, which is the incorporation of plant biomass into the soil, further enhances microbial growth, stimulates decomposer activity, and accelerates organic matter turnover [9,10]. Oats (*Avena sativa* L.) and vetch (*Vicia sativa* L.) are particularly promising among various crop combinations. The oats establish rapidly and suppress weeds, while vetch, through symbiosis with nitrogen-fixing bacteria, increases nitrogen availability and improves carbon-to-nitrogen balance in soil [11,12]. When used together, they have been shown to enrich arbuscular mycorrhizal fungi (AMF), reduce pathogenic taxa such as *Fusarium*, and support beneficial genera like *Mortierella* and *Trichoderma* [5,13,14].

Fungi play key roles in agroecosystems, contributing to nutrient cycling, soil structure stabilization, and plant health. Dominant fungal phyla such as Ascomycota, Basidiomycota, and Mortierellomycota participate in organic matter decomposition, symbiotic nutrient exchange, and biological control of pathogens [15]. Advances in high-throughput sequencing now enable detailed analysis of fungal community diversity and structure, using alpha diversity indices (e.g., Shannon, Simpson) and beta diversity metrics (e.g., Bray–Curtis, UniFrac) to compare communities across management systems [16]. Despite increasing evidence that diversified cropping enhances soil microbiomes, there is still limited field-based research on how intercropping and green manuring jointly influence fungal communities under real-world agricultural conditions. Oats provide vigorous vegetative growth and weed suppression, while vetch acts as a nitrogen-fixing species that enhances rhizosphere nutrient pools and stimulates microbial recruitment through symbiotic interactions and root exudates [17]. The contrasting root chemistries and ecological strategies of cereals and legumes have been shown to alter microbial assembly patterns by driving nutrient turnover, organic carbon inputs, and shifts in the balance between saprotrophic and symbiotic taxa [18]. These combined traits suggest that oat–vetch systems may influence fungal diversity in ways that differ from other intercrop combinations. Oat–vetch intercropping is known to improve soil fertility and boost microbial activity [19,20], its specific effects on fungal diversity, functional guilds, and community structure remain insufficiently understood. Given the central role of fungi in nutrient cycling, plant–soil interactions, and organic matter decomposition, understanding their response to cereal–legume diversification is essential for linking agricultural practices with soil ecological functioning. We hypothesize that oat–vetch intercropping, followed by its application as green manure, will enhance fungal diversity and restructure fungal community composition by enriching saprotrophic and beneficial symbiotic taxa through increased rhizodeposition and nutrient turnover.

This study addresses that gap by investigating the effects of oat–vetch intercropping, with and without green manure application, on rhizosphere fungal diversity and community composition. Using high-throughput ITS2 sequencing and multivariate analyses, we assess how these management practices shape fungal assemblages and their potential implications for sustainable soil health and crop productivity.

## 2. Materials and Methods

### 2.1. Study Site and Experimental Design

The field experiment was conducted in 2022 at the experimental fields of the Agricultural University of Plovdiv, Bulgaria (42.1335° N, 24.7498° E). The soil in the area is mainly alluvial meadow (mollic fluvisol, according to the FAO’s classification). A randomized block design with three treatments with three replicates each was applied: control field (uncultivated), oats (*Avena sativa* L., cultivar Max), and oats–vetch intercropping (*Vicia sativa* L. cultivar Obrazets 666). Each replicate was located on a plot of 35 m^2^, 105 m^2^ per treatment. The oat seeds were sown in mid-March: oats at 165 kg/ha, while in the intercropping, the seeds were planted at 215 kg/ha (3:1, vetch:oats). The fertilization was performed in the previous autumn with 200 kg/ha NPK (15:15:15) before basic tillage. At the end of June, the oats were harvested, while the oats–vetch intercropping mixture was incorporated into the soil at 30 cm as green manure. Rhizosphere soil samples were taken from each of the three replicates from 0 to 20 cm depth at three sampling times were conducted: S1—before sowing (bulk control soil), S2–S4—rhizosphere at oat ripening: S2 (bulk soil), S3 (oat rhizosphere), S4 (oat–vetch rhizosphere) and S5–S6—three months after harvest: S5 (post-green manure soil), S6 (autumn control soil) (Figure 1). The control field consisted of non-intercropped, conventionally tilled soil without organic inputs or green manure, receiving only baseline fertilization consistent with regional practice. The sampling plants in the corresponding replicate were randomly selected at each sampling time. The resulting samples were analyzed using high-throughput amplicon sequencing. Standard agronomic practices were applied equally across all treatments.

### 2.2. Soil Collection, Storage and Control

Soil sampling was performed using sterile stainless-steel trowels at a depth of 0–20 cm from each plot replicate (35 m^2^), following a randomized zig-zag pattern to obtain representative rhizosphere soil [17]. For rhizosphere sampling, plants were carefully uprooted using sterile gloves; loosely adhering soil was removed, and soil tightly attached to the roots was collected. Approximately 500 g of soil per replicate was obtained by pooling five subsamples into a composite sample. After each sample collection, tools were disinfected with 70% ethanol and flame-sterilized to avoid cross-contamination among treatments. Samples were transferred into sterile zip-lock bags, stored in insulated boxes on ice, and transported to the laboratory within 1 h after collection. Upon arrival, all samples were immediately homogenized and passed through a sterile 2 mm sieve, aliquoted, and stored at –20 °C until DNA extraction to prevent nucleic acid degradation.

### 2.3. Physicochemical Properties

Samples were homogenized in deionized H_2_O (1:10, *w*/*v*), shaken (25 min), and left to settle for 15 min, followed by measuring pH and electrical conductivity (EC) using laboratory pH-EC-meter [21]. Total nitrogen was assessed using the Kjeldahl method. Available phosphorus was assessed by the method of ammonium molybdate [22]. Soil potassium was measured after extracting it using 1N HCl and consecutive measurement at PFP-7 flame photometer (Jenway Industries Co. Ltd., Pinner, UK) [19]. Total organic carbon (TOC) was assessed using the potassium dichromate method [23,24,25].

### 2.4. DNA Extraction

Soil samples were homogenized and passed through a sterile 2 mm sieve immediately after sampling and prior to physicochemical analyses to remove stones, coarse organic debris, and root fragments [17]. Total DNA was extracted from 0.5 g of rhizosphere soil using the DNeasy PowerSoil Kit (QIAGEN, Hilden, Germany), following the manufacturer’s protocol with minor optimizations for high-organic matter soils. To improve DNA yield and reduce humic acid interference, the following optimisation steps were included: addition of Solution C1, samples were incubated at 65 °C for 10 min to improve cell disruption, and solubilisation of humic substances. DNA was eluted in two sequential 50 µL elutions with Solution C6 to maximize recovery. DNA concentration was measured using a Qubit 4 Fluorometer (Thermo Fisher Scientific, Waltham, MA, USA). DNA purity was confirmed by a A260/280 ratio of 1.80–2.00, and integrity was verified by 1% agarose gel electrophoresis, ensuring the presence of high-quality, intact DNA suitable for downstream amplicon sequencing.

### 2.5. Amplicon Sequencing and Library Preparation of Fungal Communities

Fungal community profiling was performed by amplifying the ITS2 region using primer pair ITS3 (5′-GCATCGATGAAGAACGCAGC-3′) and ITS4 (5′-TCCTCCGCTTATTGATATGC-3′), both modified with Illumina overhang adapters for compatibility with the Nextera XT workflow. For each soil DNA extract, three independent PCR reactions were performed to minimize stochastic amplification bias. Each 25 µL PCR contained 12.5 µL Phusion High-Fidelity PCR Master Mix (New England Biolabs, Ipswich, MA, USA), 0.5 µM of each primer, and 10–20 ng template DNA. PCR cycling was as follows: initial denaturation at 95 °C for 30 s; 30 cycles of 95 °C for 10 s, 55 °C for 30 s, 72 °C for 60 s; final extension at 72 °C for 5 min. PCR success and amplicon specificity were verified by electrophoresis on 1.5% agarose gels. Purified amplicons were quantified using the Qubit dsDNA High-Sensitivity assay and inspected on an Agilent Bioanalyzer (DNA 1000 kit) to verify fragment size (~300–450 bp) and absence of primer dimers. Samples failing to meet minimum concentration or size-quality criteria were re-amplified or excluded. Indexing was performed in a second PCR (8 cycles) using the Nextera XT dual-index adapters following Illumina protocols, followed by a second bead-based cleanup. Indexed libraries were re-quantified and verified before pooling. To avoid sequencing bias due to uneven read distribution among samples, each library was normalized to an equimolar concentration of 4 nM prior to pooling. The final library pool was denatured and diluted according to MiSeq specifications and supplemented with 15% PhiX Control v3 to enhance run diversity. Sequencing was performed on an Illumina MiSeq platform (2 × 300 bp, paired-end chemistry) using the MiSeq Reagent Kit v3. The run yielded sufficient sequencing depth across all samples, and quality filtering thresholds (Q ≥ 30 for >85% of bases) were met. Raw FASTQ files were demultiplexed in Illumina BaseSpace and transferred to QIIME2 for downstream processing.

### 2.6. Bioinformatics and Taxonomic Assignment

Raw paired-end sequences were processed using QIIME2 v2023.2 [26,27,28]. After demultiplexing, sequences were quality-filtered (minimum Phred score ≥ 30) and denoised using the DADA2 plugin to correct sequencing errors, remove chimeras, and generate amplicon sequence variants (ASVs) [29] (Appendix A). The resulting ASV table was curated by removing singleton ASVs, chloroplast and mitochondrial reads, as well as all non-fungal sequences. Only high-quality fungal ASVs were retained for downstream analyses. Taxonomic assignment was performed using a Naïve Bayes classifier trained on the UNITE fungal ITS reference database (version 10.0, release 19 February 2025) [30]. A 97% similarity threshold was used for species-level assignment. Taxonomic classifications required a minimum confidence of 0.70; sequences below this threshold were assigned to the highest supported rank.

### 2.7. Diversity and Bioinformatic Analyses

Alpha diversity indices (Observed ASVs, Shannon, Simpson, and Chao1) were calculated in QIIME2 using the qiime diversity plugin [26]. To account for sequencing depth variability, all samples were rarefied to 21,540 reads, corresponding to the minimum post-filtering read depth across the dataset. Rarefaction curves were generated using the qiime diversity alpha-rarefaction command.This rarefaction level ensured that alpha diversity values were directly comparable among treatments and not influenced by uneven sampling effort [26].

Beta diversity was assessed using Bray–Curtis dissimilarities and weighted UniFrac distances [16], followed by ordination analyses including Principal Coordinates Analysis (PCoA) to visualize treatment-specific community patterns. Differences in fungal community composition among treatments were evaluated using PERMANOVA (999 permutations, *p* < 0.05) [31].

Relative abundance profiles at the phylum and genus levels were generated from total-sum-scaled (TSS) normalized ASV counts. Heatmaps and rank–abundance curves were created in R (v.4.3.1) using the phyloseq and vegan packages [30,31,32]. Rarefaction curves were constructed to verify that sequencing depth was sufficient to capture fungal diversity across treatments [26]. Hierarchical clustering based on UniFrac distances was used to examine community similarity and treatment-dependent grouping [16].

### 2.8. Statistical Analysis

Statistical analyses and graphics were performed in R v4.3.1 (R Foundation for Statistical Computing) using the phyloseq, vegan, and ggplot2 packages [30,31,32]. Microbial diversity metrics were computed in QIIME2 v2023.2. Differences among treatments were evaluated using one-way ANOVA (*p* ≤ 0.05), followed by the Least Significant Difference (LSD) post hoc test to determine pairwise significance.

## 3. Results

### 3.1. Physicochemical Analysis

Data showed a slightly alkaline pH and relatively low ECs. The soil analysis showed low quantities of available nutrients. It was poorly stocked with available N, better supplied with accessible P, and well supplied with K (Table 1). Soil acidity in rhizosphere samples (plant treatments, S3–S5) demonstrated lower values than the controls, which we related to the root exudation and consecutive root effect to soil microbiota (Table 1). Available N concentration was improved in the intercropping case compared to the oats seeding. Soil P content was 37.8% lower in the intercropping than to the oats seeding; relating it with vetch needs of P, the enhanced absorption in mixed cropping, and with the limited P pool. Soil K was much higher in mixed cultivation than in oats seeding alone. TOC varied among treatments (Table 1). The highest TOC levels were observed in the intercropped rhizosphere and after green manuring, with 6.53 g/kg in S3 and 7.22 g/kg in S5, indicating enhanced carbon inputs from root exudation and decomposing biomass. Intermediate TOC values were recorded in S1 (5.41 g/kg), S4 (5.10 g/kg), and S6 (4.97 g/kg), while the lowest TOC occurred in the bulk oat soil at ripening (S2, 4.37 g/kg). These results show that intercropping and green manure incorporation increased soil organic carbon compared with bulk and untreated soils.

### 3.2. Relative Abundance and Taxonomic Composition

#### 3.2.1. Relative Abundance of Dominant Fungal Genera Across Soil Treatments

The composition of dominant fungal genera varied markedly across the six agroecological treatments (S1–S6), demonstrating clear effects of cropping system and seasonal stage on rhizosphere mycobiome structure. Only genera were included in the analysis, and all ASVs assigned to the same genus were summed prior to calculating relative abundances. In the untreated bulk soil before sowing (S1), the fungal community was dominated by *Fusarium*, followed by *Mortierella* and *Aureobasidium* (Figure 2). This profile is characteristic of unplanted alluvial-meadow soils, where opportunistic saprotrophs and potential phytopathogens prevail under limited root influence and low organic input. The relatively high abundance of *Fusarium* indicates the persistence of resident soil-borne taxa prior to crop establishment, while *Mortierella* and *Aureobasidium* likely reflect baseline saprotrophic activity associated with native soil organic matter. S1 represents the baseline fungal community in the absence of plants.

S2 (bulk soil at oat ripening stage) differs from S1 despite also being bulk soil. In S2, the relative abundance of *Aureobasidium* increases, while *Fusarium* and *Mortierella* decrease compared with S1 (Figure 2). This indicates that the presence and development of oats (*Avena sativa*) exert an indirect influence on bulk soil fungal communities, even without direct rhizosphere contact. At the oat ripening stage in bulk soil (S2), a clear shift in fungal community composition was observed compared with S1 (Figure 2). This change suggests an indirect crop effect on bulk soil fungal communities, likely driven by altered microclimatic conditions, nutrient redistribution, and early residue inputs during crop development, despite the absence of direct rhizosphere influence.

In the rhizosphere of sole-oat cultivation (S3), *Fusarium* became more prominent, co-occurring with increased abundances of *Mortierella* and *Cystofilobasidium*. This pattern is consistent with root-driven enrichment of both pathogenic and saprotrophic genera.

The oat–vetch intercropping rhizosphere (S4) exhibited a more balanced community profile, with notable increases in *Mortierella*, *Trichoderma*, *Cystofilobasidium*, and *Aureobasidium*. These genera are often associated with beneficial functions such as decomposition, nutrient cycling, and antagonism towards soilborne pathogens. The suppression of *Fusarium* relative to S3 suggests that intercropping may shift fungal community structure toward more functionally advantageous taxa.

Green manuring of the intercropped biomass (S5) resulted in the highest overall diversity and evenness among genera. *Mortierella* remained highly abundant, accompanied by increased representation of *Cystofilobasidium*, *Solicoccozyma*, and *Trichoderma*. These genera are typically stimulated by organic matter input and active decomposition processes, indicating enhanced microbial activity during biomass incorporation.

In autumn control soils (S6), the fungal community partially resembled S1 but with reduced dominance of *Fusarium* and higher representation of *Aureobasidium* and *Solicoccozyma*, likely reflecting seasonal dynamics and changes in microhabitat conditions.

Genus-level community composition differed significantly among treatments, as confirmed by PERMANOVA (Bray–Curtis: F = 4.12, R^2^ = 0.346, *p* = 0.001). The strongest differences occurred between S1 and S5, consistent with the patterns observed in the normalized genus-level profiles. The analysis demonstrates that intercropping and green manuring promote more balanced and functionally rich fungal communities compared with monoculture and untreated soils. Beneficial saprotrophic and biocontrol-associated genera such as *Mortierella* and *Trichoderma* were consistently enriched in diversified systems, whereas *Fusarium* predominated mainly in the monoculture rhizosphere.

#### 3.2.2. Fungal Community Composition Across Treatments

Heatmap clustering (Figure 3) further confirmed that intercropping in S4 and S5 supported more balanced and complex fungal communities than monoculture and control soils. This pattern underscores that plant diversity, residue input, and soil amendments create microhabitats that sustain a broader range of fungi. The distinct patterns of taxon-specific enrichment were observed, reflecting strong effects of crop type, intercropping, green manuring, and seasonal progression. S5 exhibited the highest relative abundance across multiple fungal taxa, suggesting enhanced microbial stimulation likely due to increased organic inputs and plant diversity. Similarly, S4 showed enrichment of a specific taxa cluster, indicative of synergistic plant–microbe interactions. The mycobiome before sowing (S1) and in the autumn (S6) harbored fungal communities with lower diversity and abundance, serving as baseline profiles. Interestingly, S4 revealed a shift distinct from both S1 and S6, pointing to microbial succession driven by intercropping and combining features of both crops. S2 demonstrated selective enrichment of a few taxa and poor representation, suggesting the importance of root substances for the fungal associations, albeit less diverse than intercropped systems. These results highlight that the oats-vetch intercropping profoundly influences fungal abundance and potentially stimulates beneficial soil fungi. Seasonal effects also play a key role, with post-harvest conditions favoring taxa likely involved in saprotrophic processes.

#### 3.2.3. Rarefaction Analysis and Fungal Species Richness

Rarefaction analysis revealed significant differences in fungal species richness among the experimental treatments (Figure 4). Treatments S4 (intercropping) and S5 (green manuring) exhibited the highest richness, each harboring more than 900 observed taxa. These treatments likely benefited from active residue decomposition, enhanced organic matter availability, and diverse plant–soil interactions that promote microbial habitat heterogeneity. S3 (oats cropping) closely followed in terms of richness, supporting a comparatively diverse fungal community. The co-cultivation of multiple plant species likely increased the variety of root exudates and microhabitats, favoring the establishment and coexistence of a broader range of fungal taxa. In contrast, S1 and S2 exhibited the lowest richness, with substantially fewer observed taxa. These treatments, characterized by minimal organic inputs and simplified plant structures, restrict fungal diversity—possibly due to reduced substrate heterogeneity and more chemically homogeneous soil environments. Current results underscore the positive influence of soil management practices on the richness and complexity of rhizosphere fungal communities. Treatments like S4 and S5 create favorable conditions for sustaining a diverse and functionally active soil fungal microbiome by enhancing organic matter inputs, promoting plant diversity, and facilitating dynamic belowground interactions. Such diversity is crucial in maintaining soil health, supporting nutrient cycling, and enhancing plant resilience—key components of sustainable agricultural systems. Rarefaction curves and diversity indices showed the highest species richness in S4 and S5 (>900 observed species), followed by S3 (Figure 4).

#### 3.2.4. Rank–Abundance Analysis Reveals Community Evenness and Dominance Patterns

Rank–abundance plots illustrated in Figure 5 revealed the fungal community structures across six soil treatments (S1–S6), with species richness on the x-axis and log-transformed relative abundance on the y-axis. All curves exhibited steep initial slopes, highlighting the dominance of a few fungal taxa in each community, typical of soil microbiomes where a small subset dominates in abundance while many taxa remain rare. S1 (bulk soil before sowing) and S2 (bulk soil at ripening) exhibited the steepest declines and the shortest rightward tails in their rank–abundance curves, indicating low fungal diversity and communities dominated by a few highly abundant taxa. In contrast, S5 (intercropping combined with green manuring) and S4 (intercropping) displayed longer rightward tails, reflecting greater species richness and a more even distribution among low-abundance taxa. S3 and S6 showed intermediate patterns, suggesting moderate diversity and community evenness. The results align with alpha diversity metrics, affirming that diversified practices such as intercropping and green manuring promote richer and more balanced fungal communities than monoculture or untreated conditions.

### 3.3. Alpha Diversity

Alpha diversity indices (Shannon, Simpson, and Chao1) demonstrated significant treatment differences (Figure 6). Statistical differences among treatments are clearly indicated through pairwise comparisons, and different lowercase letters (a–d) denote significant differences (*p* < 0.05). Observed richness varied substantially across the soil treatments. S4 (rhizosphere intercropping at ripening) and S5 (three months after green manuring) exhibited the highest richness values (>900 ASVs), indicating a larger number of detected fungal taxa. Intermediate richness was recorded in S3 (oat rhizosphere) and S6 (rhizosphere of oats at milky stage), while the lowest richness occurred in S1 (control soil) and S2 (bulk oat soil), where fewer ASVs were detected. The treatments S3, S4, and S6 exhibited the highest Shannon diversity index values (Figure 6A). It belonged to the same statistical group (group a), indicating significantly greater species richness and evenness than the other treatments. S5 displayed intermediate diversity (group b), while the lowest diversity was observed in S1 and S2 (group c), with S2 showing the lowest overall Shannon index.

Evenness patterns were reflected primarily in the Shannon and Simpson indices. Treatments S3, S4, and S6 showed higher evenness, with taxa more uniformly distributed across the community. S5 displayed moderate evenness, whereas S1 and S2 exhibited the lowest evenness, reflecting dominance by a smaller subset of taxa. The Simpson index in Figure 6B, reflecting species dominance and community evenness, showed that S3, S4, and S6 again formed the highest group (group a), with S5 slightly lower (group b). In contrast, S1 and S2 had significantly lower Simpson values, clustering into distinct c and d groups. These results suggest that fungal communities in monoculture or conventionally managed soils (S1 and S2) were less even and likely dominated by a few fungal taxa, potentially including phytopathogens. The consistent clustering of treatments S3, S4, and S6 in the highest diversity groups across both indices highlighted the beneficial effects of intercropping and rhizosphere. These agroecological strategies promote the development of more diverse and functionally balanced fungal communities in the rhizosphere, essential for maintaining soil health, enhancing nutrient cycling, and supporting sustainable crop productivity. In contrast, simplified systems such as sole oats cultivation were associated with reduced microbial complexity and resilience. These findings underscore the importance of biodiversity-based management in promoting stable and multifunctional soil ecosystems.

The Chao1 index in Figure 6C revealed significant differences in fungal richness across treatments. The highest richness was observed in S4 and S5, indicating that post-harvest conditions and green manuring enhance fungal diversity. In contrast, S1 and S2 showed the lowest richness, suggesting limited microbial stimulation. Intermediate values in S3 and S6 reflect the moderate effect of intercropping. These results highlight the positive impact of diversified cropping on soil fungal communities. Richness, evenness, and diversity indices consistently differentiated the treatments, showing clear variation in fungal community structure across the management systems.

### 3.4. Beta Diversity

Beta diversity was assessed through pairwise dissimilarity analysis among fungal communities derived from six distinct field treatments (S1–S6), visualized as a triangular heatmap (Figure 7). The matrix represents dissimilarity values (e.g., Bray–Curtis distances), with accompanying standard deviations in parentheses, and is color-coded from red (low dissimilarity) to yellow (high dissimilarity) to illustrate the degree of taxonomic divergence between sample pairs. The highest dissimilarities were comparing the fungal community structure before sowing (S1) and all other treatments, with particularly elevated values between S1 and S2 (1.788), S1 and S3 (1.719), and S1 and S5 (1.653). These results suggest the importance of plant roots as a driving factor for the shape of rhizosphere microbiome. The fungal community in S1 (soil before sowing), was compositionally distinct from those influenced by cropping—S4, S3, and S5. This divergence may reflect changes in root exudation, nutrient availability, or plant–microbe interactions driven by agricultural inputs and plant diversity. Moderate levels of dissimilarity were observed among the treatment combinations S3–S4 (0.844), S4–S5 (0.835), and S2–S3 (0.663), indicating partial overlap in community structure. These patterns likely arise from transitional microbial communities formed through plant residue decomposition (S5) or shared rhizosphere effects in intercropping systems. The lowest dissimilarity values were recorded between S2 and S4 (0.335) and S2 and S5 (0.457), suggesting higher similarity in community composition, possibly due to shared fungal taxa adapted to similar soil or cropping conditions. Fungal consortium of S6 displayed intermediate levels of dissimilarity relative to other treatments (e.g., S6–S1 = 1.384; S6–S3 = 1.236), reflecting its role as a reference background community with partial taxonomic overlap with untreated and managed soils.

Beta diversity analyses revealed clear separation among treatments, indicating strong effects of intercropping and green manuring on fungal community composition. Global PERMANOVA confirmed significant differences across all treatments (Bray–Curtis: F = 4.12, R^2^ = 0.346, *p* = 0.001; weighted UniFrac: F = 3.57, R^2^ = 0.308, *p* = 0.001; unweighted UniFrac: F = 4.89, R^2^ = 0.372, *p* = 0.001). Pairwise PERMANOVA results (Appendix A) further supported these patterns, with significant differences observed between most treatment combinations. Notably, fungal communities differed significantly between intercropping and green manure (F = 1.470, R^2^ = 0.062, *p* = 0.001), monocrop oat and green manure (F = 1.829, R^2^ = 0.078, *p* = 0.001), and monocrop vetch and green manure (F = 1.521, R^2^ = 0.071, *p* = 0.003). The comparison between intercropping and monocrop oat also showed a significant difference (F = 0.000, R^2^ = 0.000, *p* = 0.001), indicating strong divergence between these systems. Only the comparison between intercropping and monocrop vetch showed a weaker but still significant separation (F = 1.234, R^2^ = 0.055, *p* = 0.002). Together, these results demonstrate that crop diversification and organic residue incorporation significantly reshape fungal community structure.

### 3.5. Principal Component Analysis (PCA)

PCA in Figure 8 shows compositional differentiation among fungal communities across treatments. The first two principal components explained a cumulative 51.77% of the total variance, with PC1 (32.64%) capturing significant differences driven by field practices. Samples S4 and S5 were positioned furthest apart from the others along PC1 and PC2, reflecting substantial divergence in community composition likely associated with enhanced organic matter inputs and microbial turnover. In contrast, eukaryotic community in the bulk, S1 (before sowing), S2 (during ripening of oats), and S6 (in autumn) clustered more closely, suggesting relatively similar community structures. S3 (oats at ripening stage) occupied an intermediate position, indicating a transitional microbiome assembly influenced by crop diversity and soil legacy. These results indicate that diversified agroecological practices drive unique ecological dynamics in fungal communities, potentially affecting soil functionality and resilience.

### 3.6. Non-Metric Multidimensional Scaling (NMDS)

The analysis in Figure 9, based on ASV-level fungal community profiles, reveals clear separation among S1 to S6 plots, indicating compositional divergence across field conditions. S1 and S2 clustered distantly from the rest, reinforcing the distinctiveness of fungal communities under simplified or conventional systems. In contrast, S3, S4, S5, and S6 grouped more closely, suggesting a shared community structure likely shaped by higher microbial activity, diverse root exudates, and increased substrate heterogeneity. The stress value of 0 further confirmed the robustness of the NMDS ordination, indicating a high-quality representation of community dissimilarities.

UniFrac clustering in Figure 10 confirmed that soil cultivation practices significantly shape fungal communities. S1, dominated by Ascomycota, showed limited functional diversity, while S3–S5 supported broader taxonomic and phylogenetic profiles. NMDS analysis (stress = 0) confirmed distinct groupings based on management practices, highlighting the ecological impacts of intercropping and green manure. The weighted UniFrac-based clustering, combined with phylum-level taxonomic profiles, revealed distinct shifts in fungal community structure driven by field management practices. Before sowing (S1), the fungal community was the most phylogenetically distinct, dominated almost exclusively by Ascomycota, suggesting reduced diversity and limited functional redundancy under conventional conditions.

## 4. Discussion

Intercropping also increased soil EC compared with sole oats cultivation, reflecting higher ionic availability in the rhizosphere. This effect is likely driven by the activity of vetch roots and their nitrogen-fixing symbionts, which release organic acids and soluble ions that mobilize nutrients. At the same time, the more diverse and active microbial community under intercropping accelerates mineralization and decomposition of residues, enhancing the release of ammonium, nitrate, phosphate, and potassium into the soil solution [29]. The combined influence of legume rhizodeposition and intensified microbial processes therefore explains the observed EC increase and indicates a more dynamic nutrient turnover in intercropped soils. The TOC patterns observed in our study corroborate the positive effect of diversified management on soil carbon dynamics. The elevated TOC in the intercropped rhizosphere (S3, 6.53 g/kg) and after green manuring (S5, 7.22 g/kg) suggests increased inputs of root-derived carbon and rapid microbial turnover. Legume–cereal intercropping is known to enhance rhizodeposition through the release of organic acids, polysaccharides, and nitrogen-rich exudates, which stimulate microbial activity and accelerate carbon cycling [11,17]. Previous studies have demonstrated that organic amendments and diversified cropping systems increase labile carbon pools and promote microbial biomass accumulation due to enhanced substrate availability [9,10]. In contrast, the lower TOC detected in the bulk oat soil (S2, 4.37 g/kg) reflects limited organic matter inputs and reduced microbial stimulation typical of monoculture systems. Similar findings have been reported in conventional single-species crops, where restricted root diversity and minimal residue input result in slower carbon accumulation [6]. TOC data support the conclusion that intercropping and green manuring improve soil carbon sequestration and promote biologically active rhizosphere conditions.

High-throughput sequencing data (Appendix A) revealed treatment-dependent shifts in fungal community composition. Ascomycota dominated across all samples, particularly in control soils (S1, S4, S6), exceeding 75% of total abundance—consistent with reports that this phylum prevails in nutrient-limited, undisturbed soils due to its adaptability and saprotrophic potential [15,31]. In the oat monoculture (S2), Ascomycota slightly declined, while Fungi_phy_Incertae_sedis and Mortierellomycota increased, likely reflecting root exudate changes and carbon input patterns associated with single-species cultivation [1,10]. Intercropping (S3) supported higher fungal diversity, with greater proportions of Basidiomycota, Mortierellomycota, and Glomeromycota, indicating enhanced symbiotic and decomposer activity driven by legume-associated nitrogen fixation and rhizodeposition [5,7,11]. Post-harvest soils (S4) exhibited similar profiles with additional increases in Chytridiomycota and Glomeromycota, suggesting microbial succession following residue incorporation [9]. The strongest shift occurred in S5 (intercropping + green manuring), where Ascomycota abundance declined while Basidiomycota and Mortierellomycota dominated, reflecting accelerated organic matter turnover and saprotrophic stimulation [13,17]. Control soil S6 resembled S1, maintaining Ascomycota-dominated communities typical of conventional or unmanaged soils. Overall, crop diversification and green manuring enhanced fungal diversity and functional potential, corroborating previous findings that intercropping fosters beneficial taxa such as *Mortierella*, *Glomus*, and *Trichoderma*, while reducing *Fusarium* [33,34]. Such diversified systems promote resilient fungal assemblages essential for nutrient cycling, soil structure, and sustainable ecosystem productivity [2,3].

The present investigation demonstrates that field management strategies and plant diversity strongly influence rhizosphere fungal communities’ structure, richness, and functional complexity (Figure 2). Fungal diversity was markedly higher in S3 and S5, likely driven by the synergistic effects of crop diversification and organic amendments, compared with the simpler management regimes of S2, S1, and S6. Increased abundance of *Mortierella* and Glomeromycota in intercropped plots suggests that these practices may support beneficial fungi, potentially suppressing *Fusarium*, aligning with findings of Vukicevich et al. (2016), Pużyńska et al. (2021), and Qu et al. (2022) [5,35,36] (Figure 3).

This result was confirmed by a circular phylogenetic tree based on the top 100 most abundant fungal genera (Appendix A), which revealed distinct taxonomic and evolutionary patterns among treatments. Most genera were affiliated with Ascomycota, confirming its ecological dominance and broad adaptive potential in rhizosphere soils [15,33]. Basidiomycota and Mortierellomycota were also well represented, followed by minor contributions from Mucoromycota, Glomeromycota, and Chytridiomycota, reflecting both saprotrophic and symbiotic functional groups. Treatments combining intercropping and green manuring (S5) or intercropping alone (S3) exhibited markedly higher phylogenetic diversity, evidenced by multiple deep-branching clades and broader taxonomic dispersion. These findings suggest that crop diversification and organic residue incorporation generate heterogeneous microhabitats that foster the coexistence of evolutionarily distinct fungal lineages [5,9,17]. In contrast, control soils (S1, S6) showed more conserved phylogenetic architectures dominated by Ascomycota, indicating reduced evolutionary divergence under uniform and nutrient-limited conditions [1]. Genera such as *Cladosporium*, *Penicillium*, and *Trichoderma* were consistently detected across all treatments, forming a core fungal microbiome known for saprotrophic activity, biocontrol potential, and contribution to nutrient turnover [32,34]. Their persistence highlights functional robustness within soil fungal communities irrespective of management regime.

Alpha diversity metrics revealed apparent shifts in fungal community structure in response to different cropping systems and soil management strategies (Figure 6). Treatments S4 and S5, which highlighted the role of intercropping followed by the green manuring, consistently showed the highest values across all three diversity indices—Shannon (6.32–6.37), Simpson (0.962–0.965), and Chao1 (961–1009)—indicating the presence of rich and well-balanced fungal communities. These systems likely provide continuous inputs of organic matter and microhabitat diversity, enhancing niche availability and favoring both dominant and rare fungal taxa. Previous studies have shown that organic amendments stimulate fungal richness by supplying decomposable substrates and supporting functional microorganisms such as saprotrophs and mycorrhizae [9,30]. In contrast, S1 (control) and S2 (oat monoculture) exhibited the lowest Shannon values (<4.9) and relatively lower Chao1 estimates (<790), suggesting that the absence of plant diversity and minimal organic inputs restrict fungal complexity (Figure 6A). The reduced Simpson index in S1 (0.884) indicates dominance by a limited number of fungal taxa, likely outcompeting others under nutrient-poor and structurally simplified soil conditions, where root systems and external nutrient inputs were absent (Figure 6B). Such environments select for stress-tolerant or opportunistic fungi, potentially narrowing the functional breadth of the community [33]. Intermediate diversity was observed in S3 and S6, with Shannon values of 5.89 and 5.62 and Chao1 richness estimates of 828 and 861, respectively. These values suggest moderate gains in fungal diversity under intercropping, likely due to root exudate diversification and the complementary effects of legumes in improving nitrogen availability. S6, although unmanaged, exhibited a relatively high Simpson index (0.933), indicating a fairly even community likely influenced by seasonal shifts and spontaneous vegetation. The Chao1 index in Figure 6C offered a unique insight into community richness by estimating undetected taxa, highlighting a broader diversity pool in S4 and S5. This index is susceptible to rare species, which may contribute to ecological functions such as pathogen suppression or nutrient cycling [35,36]. Chao1 with Shannon and Simpson confirms that organic and diversified treatments do not simply increase richness, but also support a more balanced distribution of taxa. These patterns reinforce the ecological value of practices, especially intercropping and green manuring, for fostering diverse and functionally rich fungal communities in the rhizosphere. Such communities are critical for maintaining soil health, nutrient dynamics, and agroecosystem resilience under sustainable farming systems.

Lower beta diversity values observed between some treatment pairs, such as S2 and S4 (0.335) as well as S2 and S5 (0.457), suggesting partial convergence of fungal community composition, possibly driven by similar ecological conditions or shared plant residue inputs (Figure 7). The resemblance between bulk eukaryotic microbiome during ripening (S2) and intercropping (S4) likely reflects the dominance of saprotrophic fungi actively colonizing decaying plant material. In contrast, bulk eukaryotic assemblage in autumn (S6) showed intermediate dissimilarity with all other treatments, indicating that even unmanaged soils are influenced by seasonal dynamics and microsite heterogeneity within the field environment. Although ANOSIM results (R = −0.0019, *p* = 0.532) did not reveal statistically significant separation among treatments, the observed patterns in beta diversity and UniFrac clustering highlight that management intensity and plant diversity can strongly modulate rhizosphere fungal assembly. These practices enhance ecological niche partitioning, improve community stability, and promote resilience to environmental fluctuations [37,38]. The trends indicate that diversified cropping systems favor richer and more functionally balanced fungal networks, supporting rare taxa and fostering long-term soil health. Pairwise dissimilarity analyses and ordination methods (PCA, NMDS; Figure 8 and Figure 9) revealed strong differentiation between conventional controls (S1, S2) and diversified treatments (S3–S5), and this is consistent with [16]. Weighted UniFrac clustering (Figure 10) confirmed that intercropping with or without green manure shifted communities from Ascomycota-dominated profiles toward more functionally diverse fungal assemblages. Although overall ANOSIM results indicated high within-group variability, consistent trends showed that management intensity and plant diversity significantly shape fungal community composition and phylogenetic breadth. PERMANOVA and UniFrac-based clustering confirmed that diversified systems foster a wider range of phyla, including *Biasidiomycota*, *Mortierellomycota*, and *Glomeromycota*. In the bulk fungal soils (S1, S2, and S6) were dominated by Ascomycota and showed low functional diversity (Appendix A). In contrast, S4 and S5 hosted complex fungal communities, including rare or unclassified taxa such as Zoopagomycota and Fungi_phy_Incertae_sedis, suggesting that agroecological inputs can uncover hidden microbial potential.

Figure 11 presents β-diversity distances (weighted and unweighted UniFrac) between bulk control soils and rhizosphere soils. The greatest dissimilarity was observed in S5 (intercropping + green manuring) and S2 (oat monoculture), indicating substantial shifts in the fungal community composition under distinct management regimes. These pronounced differences suggest that both increased management intensity and the incorporation of plant residues significantly restructure the soil mycobiome. Moderate divergence in S3 (oat–vetch intercropping) and S4 (post-harvest stage) reflects transitional community responses influenced by interspecific plant interactions and residual organic inputs. In contrast, the lower variability observed in S1 (control) points to the natural background heterogeneity of the rhizosphere. These results demonstrate that agroecological practices promoting plant diversity and organic matter input drive significant modifications in fungal community structure and β-diversity, corroborating previous findings by Qu et al. (2022) and Lori et al. (2017) that link soil management intensity to microbial community differentiation [5,9].

The enrichment of saprotrophic and symbiotic fungal groups under intercropping and green manuring likely reflects shifts in rhizodeposition and nutrient cycling. Legume-derived nitrogen and diverse root exudates can stimulate the proliferation of fungi involved in decomposition and organic nitrogen transformation, while complementary root traits promote niche partitioning and microbial recruitment. These functional inferences should be interpreted with caution because the current study is based solely on taxonomic markers. In the absence of enzyme assays or metatranscriptomic profiling, the ecological roles of specific taxa remain speculative. Additionally, the single-year design limits insights into the temporal stability of fungal responses. Future investigations integrating functional activity assays and multi-season datasets are needed to validate the mechanistic links between diversification practices and fungal-mediated soil processes [39].

In contrast, intercropping, root exudates, and green manuring (S3 and S5) promoted phylogenetically and taxonomically diverse communities, with increased representation of Basidiomycota, Mortierellomycota, Glomeromycota, and rare phyla such as Zoopagomycota and Blastocladiomycota (Appendix A). Interestingly, unclassified fungal taxa in organically managed soils suggest that agroecological practices may also foster previously understudied or novel microbial lineages. The findings revealed an additional ecological benefit of diversified systems: expanding the unexplored microbial diversity pool, which could offer new functional opportunities for soil ecosystem services.

## 5. Conclusions

This study demonstrates that oat–vetch intercropping and green manuring strongly influence the structure, richness, and composition of rhizosphere fungal communities. Diversified management practices (S3–S5) supported higher fungal diversity, more even community structures, and broader taxonomic representation compared with simplified systems (S1, S2, S6). These practices enhanced beneficial fungi such as *Mortierella*, *Glomus*, and *Trichoderma*, while limiting potential pathogens including *Fusarium*. Beta diversity and ordination analyses further showed that management intensity and crop diversity shape distinct fungal assemblages with greater functional potential. Importantly, the positive legacy of intercropping remained detectable post-harvest, indicating lasting benefits for soil microbial resilience. Collectively, the findings highlight the value of biodiversity-based practices in fostering stable and functionally rich soil mycobiomes, contributing to sustainable crop productivity and long-term agroecosystem health.

## Figures and Tables

**Figure 1 microorganisms-14-00042-f001:**
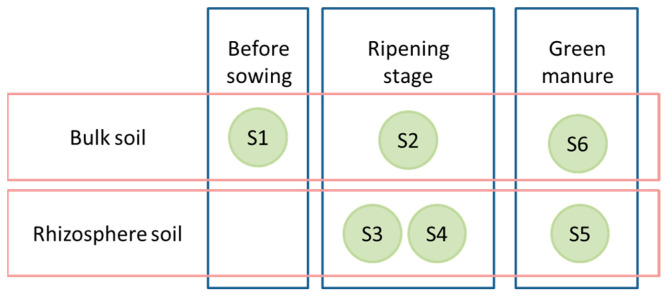
Origin of soil samples and time of collection. The soil samples of each stage were collected on the same day.

**Figure 2 microorganisms-14-00042-f002:**
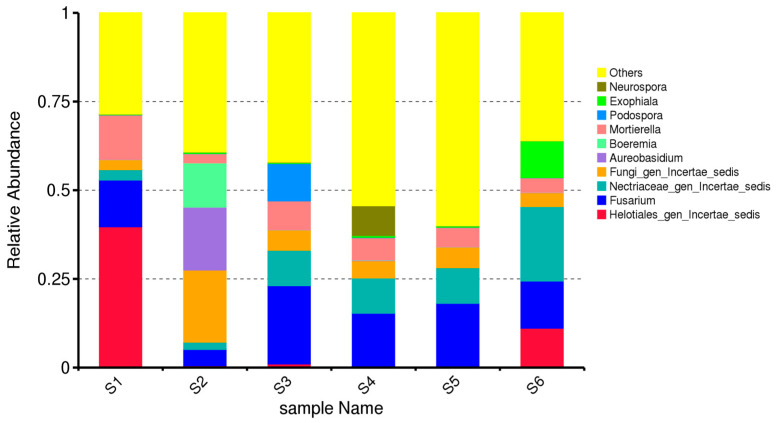
Genus-level relative abundance of fungal communities across the six soil treatments (S1–S6). The stacked bar plots display the ten most abundant fungal genera across all samples, based on total-sum-scaled (TSS) normalized ASV counts. All remaining genera with lower relative abundances are grouped under “Others”. ASVs assigned to the same genus were aggregated prior to visualization. The figure highlights treatment-specific shifts in fungal community composition among bulk soil, rhizosphere samples, and green-manured plots.

**Figure 3 microorganisms-14-00042-f003:**
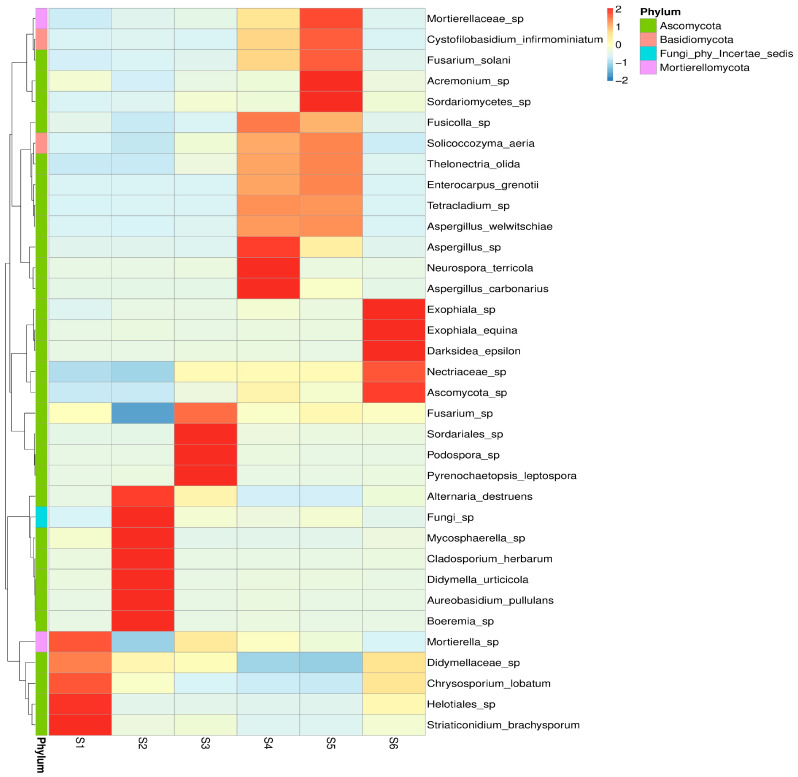
Heatmap illustrating the Z-score-standardized abundance of dominant fungal genera across the six soil treatments (S1–S6). Values range from –2 to +2, representing deviations from the mean abundance of each genus across samples (Z = (x − μ)/σ). Colour gradients represent variations in relative abundance, with red-brown tones indicating higher abundance. Clustering on the vertical axis reflects taxonomic similarity or co-occurrence patterns among fungal genera.

**Figure 4 microorganisms-14-00042-f004:**
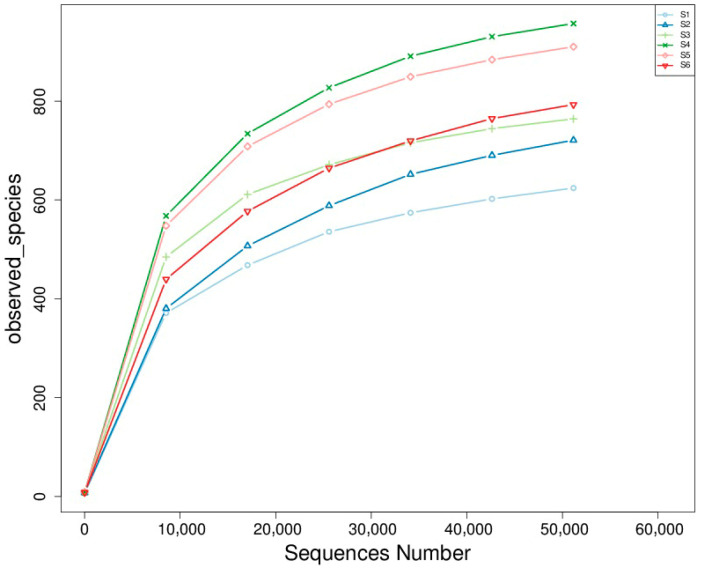
Rarefaction curves of observed fungal species across treatments. The curves show the relationship between sequencing depth (number of sequences) and the number of observed fungal species in each treatment group. The saturation trend indicates fungal communities’ richness and sampling completeness under different agroecological conditions.

**Figure 5 microorganisms-14-00042-f005:**
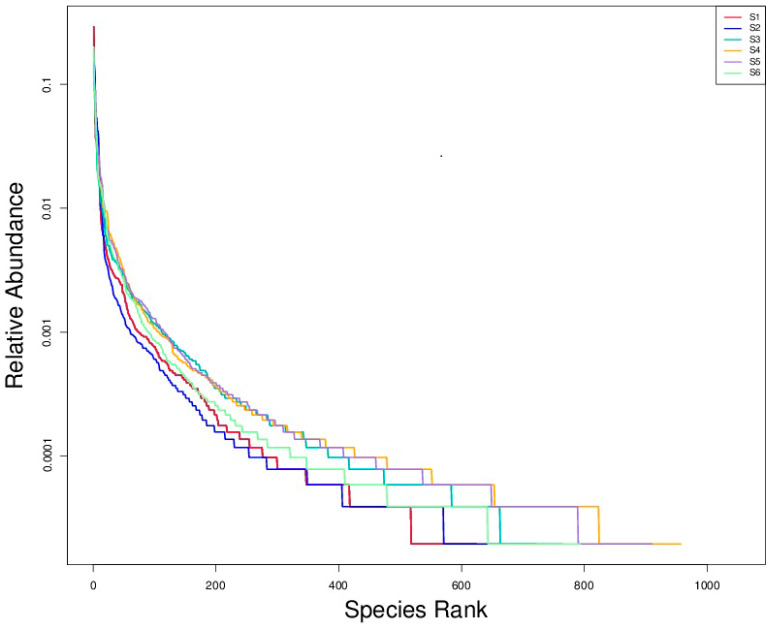
Rank–abundance curves of fungal communities under different treatments. The curves illustrate the distribution of fungal taxa based on relative abundance and rank, providing insights into community evenness and dominance patterns.

**Figure 6 microorganisms-14-00042-f006:**
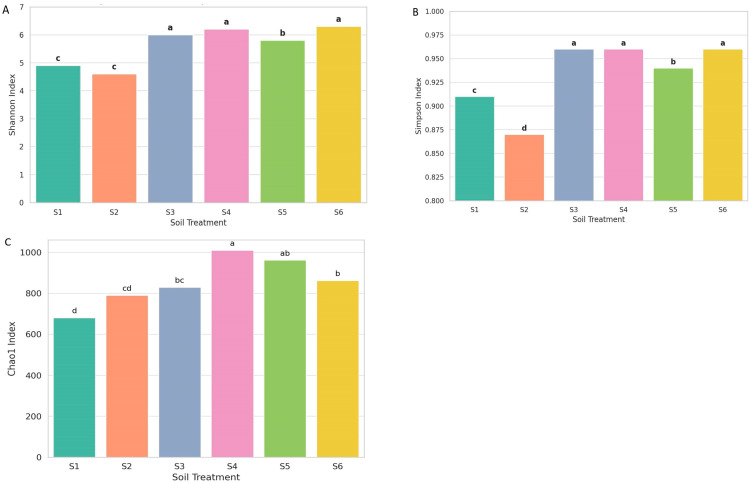
Alpha diversity of fungal communities across treatments based on Shannon diversity index (**A**), Simpson diversity index (**B**), and Chao1 diversity index (**C**). Box plots represent median values, different letters (a–d) indicate significant differences among treatments (*p* < 0.05).

**Figure 7 microorganisms-14-00042-f007:**
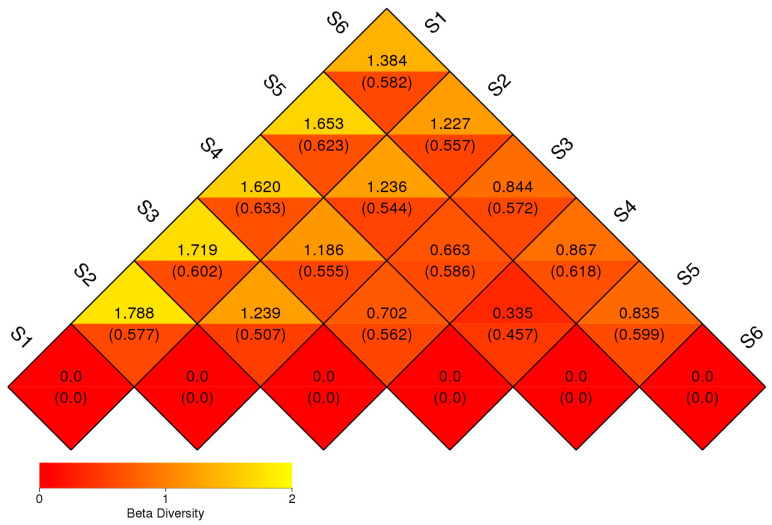
Beta diversity heatmap of fungal communities across treatments. The heat-map highlights the variation in microbial composition between the samples. The numbers inside the parentheses refer to the UniFrac-unweighted analysis, while those shown outside refer to the UniFrac-weighted analysis.

**Figure 8 microorganisms-14-00042-f008:**
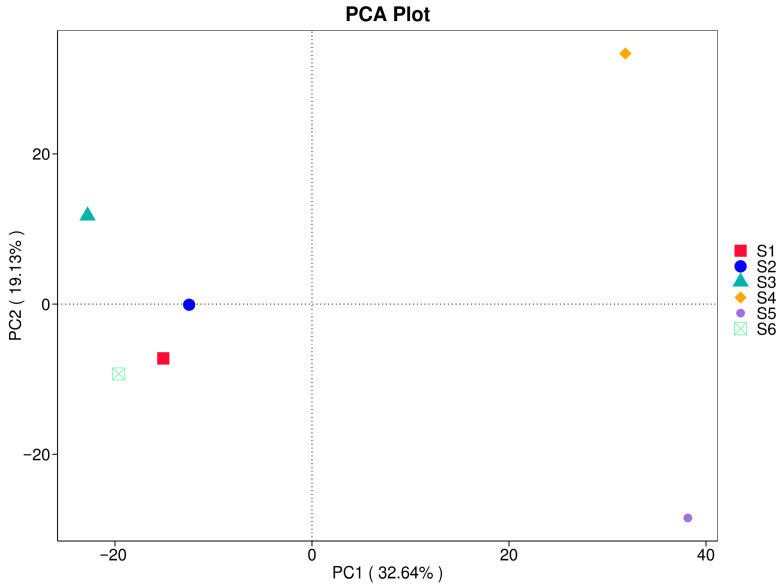
Principal component analysis (PCA) of soil fungal communities under different treatments (S1–S6).

**Figure 9 microorganisms-14-00042-f009:**
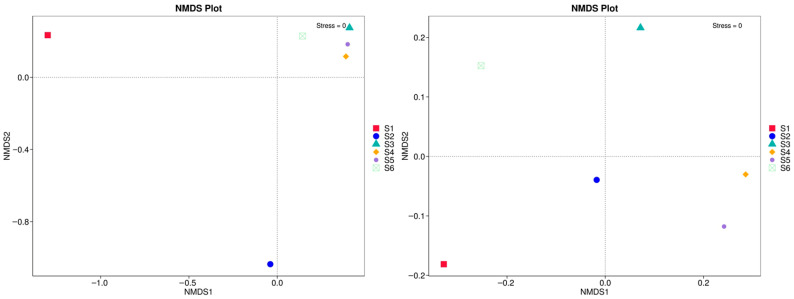
Non-metric multidimensional scaling (NMDS) plots based on fungal community composition.

**Figure 10 microorganisms-14-00042-f010:**
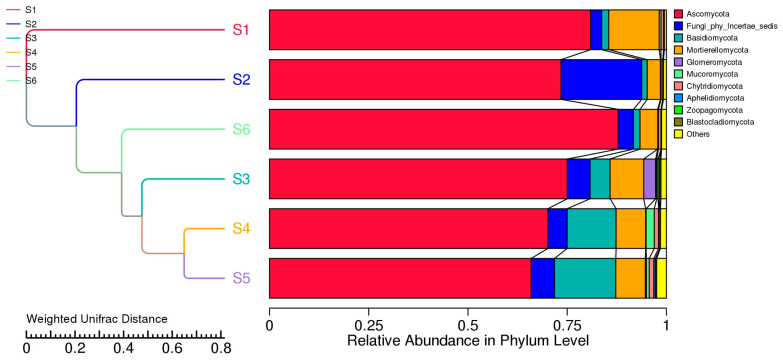
UniFrac-based clustering and relative abundance of fungal phyla. Hierarchical clustering of fungal communities based on weighted UniFrac distance (**on the left**) reveals compositional dissimilarity among the six field treatments (S1–S6), reflecting differences in fungal community structure. The accompanying stacked bar plots (**on the right**) shows the relative abundance of dominant fungal phyla in each treatment.

**Figure 11 microorganisms-14-00042-f011:**
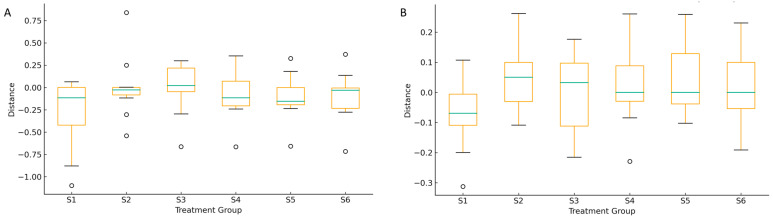
Beta diversity distances to the reference fungal community (RG1) across treatment groups (RG2–RG5) based on weighted (**A**) and unweighted (**B**) UniFrac. The boxplot illustrates the distribution of pairwise distances between fungal communities in each group and the RG1 reference group (corresponding to S1, control). Greater distances indicate higher dissimilarity in community structure, reflecting the impact of different practices.

**Table 1 microorganisms-14-00042-t001:** Physicochemical properties of the soil of studied field on a dry weight basis. Data shown the mean and LSD. Different small letters denoted existent differences between treatments.

	pH (H_2_O)	EC (μS/cm)	Available N (mg/kg)	Available P (mg/kg)	Available K (mg/kg)	TOC (g/kg)
S1	7.76 ab	189.3 a	20.5 a	2.53 b	150.2 a	5.41 b
S2	7.80 a	143.9 c	11.2 d	3.56 a	97.5 c	4.37 b
S3	7.62 c	126.1 d	16.2 c	3.77 a	105.3 c	6.53 a
S4	7.74 bc	181.7 b	19.5 ab	2.34 b	131.7 b	5.10 b
S5	7.63 d	114.8 e	17.0 bc	3.93 a	128.3 b	7.22 a
S6	7.78 ab	191.0 a	15.8 c	2.14 b	127.1 b	4.97 b

## Data Availability

The sequencing data generated in this study were deposited in the NCBI Sequence Read Archive (SRA) https://submit.ncbi.nlm.nih.gov/subs/sra/SUB15507337/overview, accessed on 1 November 2025, under BioProject accession number PRJNA1299282 (Rhizosphere Fungal Community Dynamics in Oat-Vetch Intercropping and Green Manure Systems) on 31 July 2025. Corresponding BioSample records for the individual rhizosphere soil samples are available under the following accessions: SAMN50298520, SAMN50298521, SAMN50298522, SAMN50298523, SAMN50298524, and SAMN50298525, which are publicly accessible via the NCBI portal: https://www.ncbi.nlm.nih.gov/bioproject/PRJNA1299282, accessed on 1 November 2025.

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
