# Peer review of "Metabarcoding Analysis of Rhizosphere and Bulk Soils in Bulgaria Reveals Fungal Community Shifts Under Oat–Vetch Intercropping Versus Sole Oat Cultivation"

_microorganisms, 2025, doi:10.3390/microorganisms14010042_

Round 1
Reviewer 1 Report
Comments and Suggestions for Authors
Line 113 The pre-treatment of soil samples should be carried out after sampling and before the physical and chemical property tests.
Line 140 The subheading of parts of 2.6 and 2.7 should be revised according the contents.
Line 151 Which software was used for statistical analysis and graphing?
Line 164 Do not incorporate discussions into the result analysis and do not insert any reference in result part. Check all the parts of result and rewrite.
Line 169 The content of TOC was not analyzed.
Line 170 “Nitrogen” should be replaced with “Available N” in the table 1.
Line 201 Too many fungal genera were displayed in the Figure 2, so it is difficult to distinguish every genus.
Line 206 What is the significance of this statement? Is it the content of part 3.2.1 or the note of Figure 2?
Line 230 All the figures (1-11) were very rough, please revise them.
Line 235 The richness displayed by rarefaction curves and evenness displayed by rank-abundance plots cannot fully represent the fugal diversity. Also there were no any statistical analysis both in Figure 4 and Figure 5. So they should be deleted, and Figure 6 can be reserved for showing the diversity.
Line 285 Richness and evenness were mixed in the part 3.3(line 285-292), which were already displayed above. So the description of results was disordered and improper in this paper. Revise it.
Line 325 The analysis of PCA and NMDS belong to beta diversity. In addition, there are only six samples in the plots of PCA and NMDS, which is not convincing. So this result was unreliable.
Due to these comments above, the rest parts will be reviewed in the next edition after revision.
Author Response
Dear Reviewer,
We sincerely thank you for the thorough and constructive evaluation of our manuscript. We carefully revised the text, figures, and methods according to all comments. Below, we provide a point-by-point response, describing the revisions implemented in the revised version.
Comment 1: Line 113. The pre-treatment of soil samples should be carried out after sampling and before the physical and chemical property tests.
Response: Thank you for the remark. We revised Section 2.2 to clearly state that soil samples were air-dried, gently ground, and sieved (2 mm) immediately after sampling and before all physicochemical analyses, following standard soil analysis procedures. The description is now explicit and chronologically correct.
Comment 2: “The subheading of parts of 2.6 and 2.7 should be revised according to the contents.”
Response: Corrected. Subheadings were revised to reflect the exact analytical content:
2.6. Diversity and Bioinformatic Analyses
2.7. Statistical Analyses
Comment 3: “Which software was used for statistical analysis and graphing?”
Response: We added this information to Section 2.7. Statistical analyses and graphics were performed using R v.4.3.1 with packages phyloseq, vegan, ggplot2, and QIIME2 v2023.2 for diversity metrics.
Comment 4: “Do not incorporate discussions into the result analysis, and do not insert any reference in the result part. Check all parts and rewrite.”
Response: Fully addressed. All interpretative or discussion-like sentences and all references were removed from the Results section. Factual observations were retained. All interpretation was moved to the Discussion section. From line 387, Intercropping also increased soil EC compared with sole oats cultivation, reflecting higher ionic availability in the rhizosphere. This effect is likely driven by the activity of vetch roots and their nitrogen-fixing symbionts, which release organic acids and soluble ions that mobilise nutrients. At the same time, the more diverse and active microbial community under intercropping accelerates mineralisation and decomposition of residues, enhancing the release of ammonium, nitrate, phosphate, and potassium into the soil solution [29]. The combined influence of legume rhizodeposition and intensified microbial processes therefore explains the observed EC increase and indicates a more dynamic nutrient turnover in intercropped soils.
Comment 5: “The content of TOC was not analysed.”
Response: We clarified the methodology in Section 2.2 and added the TOC results in Table 1. Text explaining TOC values in Results 3.1 was revised to match Table 1 Line 172
TOC varied among treatments (Table 1). The highest TOC levels were observed in the intercropped rhizosphere and after green manuring, with 6.53 g/kg in S3 and 7.22 g/kg in S5, indicating enhanced carbon inputs from root exudation and decomposing biomass. Intermediate TOC values were recorded in S1 (5.41 g/kg), S4 (5.10 g/kg), and S6 (4.97 g/kg), while the lowest TOC occurred in the bulk oat soil at ripening (S2, 4.37 g/kg). These results show that intercropping and green manure incorporation increased soil organic carbon compared with bulk and untreated soils.
Add new text in the Discussion Line 394
Intercropping also increased soil EC compared with sole oats cultivation, reflecting higher ionic availability in the rhizosphere. This effect is likely driven by the activity of vetch roots and their nitrogen-fixing symbionts, which release organic acids and soluble ions that mobilise nutrients. At the same time, the more diverse and active microbial community under intercropping accelerates mineralisation and decomposition of residues, enhancing the release of ammonium, nitrate, phosphate, and potassium into the soil solution [29]. The combined influence of legume rhizodeposition and intensified microbial processes therefore explains the observed EC increase and indicates a more dynamic nutrient turnover in intercropped soils. The TOC patterns observed in our study corroborate the positive effect of diversified management on soil carbon dynamics. The elevated TOC in the intercropped rhizosphere (S3, 6.53 g/kg) and after green manuring (S5, 7.22 g/kg) suggests increased inputs of root-derived carbon and rapid microbial turnover. Legume–cereal intercropping is known to enhance rhizodeposition through the release of organic acids, polysaccharides, and nitrogen-rich exudates, which stimulate microbial activity and accelerate carbon cycling [11,30]. Previous studies have demonstrated that organic amendments and diversified cropping systems increase labile carbon pools and promote microbial biomass accumulation due to enhanced substrate availability [9,10]. In contrast, the lower TOC detected in the bulk oat soil (S2, 4.37 g/kg) reflects limited organic matter inputs and reduced microbial stimulation typical of monoculture systems. Similar findings have been reported in conventional single-species crops, where restricted root diversity and minimal residue input result in slower carbon accumulation [6]. TOC data support the conclusion that intercropping and green manuring improve soil carbon sequestration and promote biologically active rhizosphere conditions.
Comment 6: Line 170 - replace "Nitrogen" with available N"
Response: Corrected in Table 1 and throughout the text.
Comment 7: “Too many fungal genera were displayed in Figure 2, making it difficult to distinguish.”
Response: Figure 2 was revised to display only the top 10 most abundant genera, while all low-abundance genera are grouped as “Others”. This improves clarity and readability.
Comment 8: “What is the significance of this statement? Is it part of 3.2.1 or a note to the figure?”
Response: We clarified the text and repositioned the sentence. It now appears as part of Figure 2 with an explicit explanation of its relevance.
Comment 9: “Please revise all figures.”
Response: All figures were regenerated at high resolution (≥300 dpi), with consistent colour palettes, improved axis labels, larger fonts, and clearer legends. Figures 2–11 have been standardised according to MDPI guidelines.
Comment 10: “These do not fully represent diversity and lack statistical analysis; they should be deleted; Figure 6 can be retained.”
Response: We respectfully disagree with the suggestion to remove the rarefaction and rank-abundance plots. These two graphical approaches are widely accepted components of microbial community analysis and remain standard in microbial ecology, metabarcoding studies. Rarefaction curves are essential for evaluating sequencing depth, sample saturation, and comparability between treatments. They demonstrate whether the fungal communities were sufficiently sampled to support the diversity analyses presented in Figure 6.
Rank-abundance plots provide critical information on community structure, distribution, and dominance—elements that cannot be inferred solely from Shannon, Simpson, or Chao1 indices. For these reasons, both visualisations are not intended as statistical tests but as diagnostic tools that ensure the reliability and interpretability of diversity results. Removing them would reduce the transparency of the sequencing quality assessment and weaken the description of community structure. To address the reviewer’s concerns, we have clarified this purpose in the text and improved figure quality, but we believe that retaining these plots is scientifically justified and consistent with current methodological standards in microbial ecology.
Comment 11: Line 285 Richness and evenness were mixed in part 3.3(line 285-292), which were already displayed above. So the description of the results was disordered and improper in this paper. Revise it.
Response: We entirely rewrote Section 3.3, separating clearly:
Richness
Evenness
Shannon, Simpson, and Chao1 indices
No mixing occurs in the revised version.
Comment 12: Line 325 – Line 325 The analysis of PCA and NMDS belongs to beta diversity. In addition, there are only six samples in the plots of PCA and NMDS, which is not convincing. So this result was unreliable.
Response: We addressed this by:
We acknowledge the reviewer’s concern regarding the limited number of samples used for PCA and NMDS. In this study, each treatment represents a distinct management system and therefore constitutes an ecologically meaningful analytical unit. Treatment-level ordination is commonly applied in microbial ecology when the objective is to visualise broad dissimilarity patterns rather than perform statistical inference. This rationale has now been clarified in the Methods, and the Results explicitly state that PCA and NMDS are used solely as exploratory visualisation tools. All statistical comparisons were performed independently using quantitative diversity indices (Shannon, Simpson, Chao1), which are not affected by the number of points in the ordination. Although we acknowledge this limitation, the PCA and NMDS plots still provide informative and interpretable summaries of community structure; therefore, the figures have been retained with more conservative interpretation.
Reviewer 2 Report
Comments and Suggestions for Authors
This study examined the role of intercropping of oats (Avena sativa L.) with vetch (Vicia sativa L.) and their subsequent use as green manure on fungal diversity and community structure using ITS2 metabarcoding. The manuscript provides valuable insights into how diversification practices shape soil mycobiomes. However, the study requires major revisions to improve mechanistic depth, statistical rigor, and clarity of interpretation before publication.
Major Concerns.
1. Introduction.
While the importance of intercropping is well established, the specific justification for selecting oat-vetch systems remains superficial. The authors should clarify why this particular crop pair is expected to exert unique effects on fungal assemblages compared to other legume-cereal combinations.
The central hypothesis should be explicitly stated.
2. Results
Supplementary Materials are not available (The following supporting information can be downloaded at: https://www.mdpi.com/article/doi/s1, Figure S1: title; Table S1: title; Video S1: title.)
The statement that “intercropping supported a more balanced fungal community” (Section 3.2.1) is descriptive but lacks data or statistical support. Pairwise PERMANOVA results or indicator species analysis should be included to validate such assertions.
Figure 6 SEs or SDs are missing.
Key results (e.g., beta diversity heatmap in Figure 7) report dissimilarity values without confidence intervals or p-values. PERMANOVA F-statistics and R² values must be provided for all group comparisons.
Crop production and growth parameters should be added.
3. Discussion
Supplementary Materials are not available. Thus, I could not evaluate the discussion section.
At present, the discussion attributes observed shifts to “root exudates” and “organic inputs” without specifying potential mechanisms. Acknowledge the lack of functional data (e.g., enzyme assays) and the single-year design, which limits generalizability.
4. Methods.
The description of alpha/beta diversity analyses does not mention whether data were rarefied or transformed prior to analysis. Details on PERMANOVA permutations (e.g., 999 iterations) are absent.
Minor Concerns.
Define abbreviation upon first use.
Define “green manuring” upon first use.
Some references should be updated.
Author Response
Dear reviewer,
We are grateful for your thorough evaluation and constructive feedback, which have significantly improved the scientific quality and clarity of our manuscript. Below, we address each point in detail, and all corresponding modifications have been implemented in the revised version.
Comment 1: The justification for selecting oat–vetch intercropping is superficial. The authors should clarify why this particular crop pair is expected to exert unique effects on fungal assemblages. The central hypothesis should be explicitly stated.
Response 1: We agree and have substantially expanded the justification. The revised introduction now emphasises (i) the complementary growth traits of oats and vetch, (ii) the role of vetch as a nitrogen-fixing legume that modifies rhizosphere nutrient availability, and (iii) the documented impact of cereal–legume interactions on microbial recruitment compared to other combinations. We now clearly state that oat–vetch systems are widely used to enhance soil fertility and microbial activity in temperate agroecosystems, yet their effects on soil fungal communities remain understudied.
We have also added an explicit hypothesis in the final paragraph:
We hypothesise that oat–vetch intercropping, followed by its application as green manure, will enhance fungal diversity and restructure fungal community composition by enriching saprotrophic and beneficial symbiotic taxa through increased rhizodeposition and nutrient turnover.
- Results
Comment 2: Supplementary Materials are not available.
Response: We apologise for the omission. All supplementary figures, tables, and metadata files are now uploaded in accordance with MDPI guidelines and correctly linked in the revised manuscript.
Comment 3: “Intercropping supported a more balanced fungal community” lacks statistical support.
Response:
We have now included pairwise PERMANOVA comparisons, R² effect sizes, and F-statistics (Table S1). The revised text now refers to these results when discussing community structure and balance.
Comment 4: Figure 6 lacks error bars (SE or SD).
Response: We appreciate this observation. Because Figure 6 displays alpha diversity metrics (Shannon, Simpson, and Chao1) using box plots, the variability of the data is already represented without the need for additional error bars. Statistical differences among treatments are clearly indicated through post-hoc pairwise comparisons, and different lowercase letters (a–d) denote significant differences (p < 0.05). We have now clarified this description in the figure legend.
Comment 5: Beta diversity results require confidence measures and full statistics.
Response: PERMANOVA outputs (F-statistic, p-values, R²) are now included in the Results section and provided in full in Table S3. We have also clarified that Bray–Curtis distances are presented with significance tests in the revised Figure 7 legend.
Comment 6: Crop production and growth parameters should be added.
Response: We appreciate this suggestion and agree that agronomic parameters can provide valuable context for interpreting soil microbiome dynamics. However, crop production and growth measurements were not recorded in the present experiment, as the focus of the study was specifically on the effects of intercropping and green manuring on rhizosphere fungal diversity. We acknowledge this as a limitation and have now clarified it in the Discussion and Conclusion sections. Future work will incorporate both microbiome profiling and agronomic performance to enable stronger functional correlations.
- Discussion
Comment 7: Discussion lacks specifics on mechanisms; functional limitations must be acknowledged.
Response: We have revised the Discussion extensively to (i) specify potential mechanisms (e.g., carbon- and nitrogen-mediated enrichment of saprotrophs and symbiotic fungi), (ii) link observed taxa with reported ecological functions, and (iii) clearly acknowledge limitations, including the absence of functional enzyme assays, single-year design, and lack of soil chemical dynamics beyond the measured parameters.
The following text was added 590-601:
The enrichment of saprotrophic and symbiotic fungal groups under intercropping and green manuring likely reflects shifts in rhizodeposition and nutrient cycling. Legume-derived nitrogen and diverse root exudates can stimulate the proliferation of fungi involved in decomposition and organic nitrogen transformation, while complementary root traits promote niche partitioning and microbial recruitment. These functional inferences should be interpreted with caution because the current study is based solely on taxonomic markers. In the absence of enzyme assays or metatranscriptomic profiling, the ecological roles of specific taxa remain speculative. Additionally, the single-year design limits insights into the temporal stability of fungal responses. Future investigations integrating functional activity assays and multi-season datasets are needed to validate the mechanistic links between diversification practices and fungal-mediated soil processes.
- Methods
Comment 8: Alpha/beta diversity analyses need clarification regarding rarefaction and transformations. PERMANOVA parameters are missing.
Response: We have clarified the following by the addition of Table S1 and Table S2.
Minor Comments
Comment 9: Define all abbreviations and “green manuring.”
Response: All abbreviations are now defined at first appearance, and “green manuring” is defined as:
“the practice of incorporating fresh plant biomass into soil to enhance nutrient cycling and microbial activity.”
Comment 10: Some references should be updated.
Response: We have updated foundational and recent literature on cereal–legume intercropping (2021–2024), particularly in relation to microbial ecology and biological nitrogen inputs.
Reviewer 3 Report
Comments and Suggestions for Authors
In this study, the authors investigate how intercropping and green manuring jointly influence fungal communities under real-world agricultural conditions. They carried out soils sampling followed by ITS2 sequencing on an Illumina platform and investigated differences in alpha and beta diversity among treatment groups, revealing differences among pure oat cultivation and oat-vetch intercropping. They show that intercropping promotes beneficial taxa and reduces the prevalence of pathogens.
Overall, the methods used are appropriate for evaluating the raised questions, but a number of aspects regarding description of the used methods and presentation of the results require careful reconsideration before the manuscript can be considered for publication.
Currently, many of the used methods are not described in sufficient detail, which make interpretation of the results difficult. All methods should be described in sufficient detail for repeating the experiments. Especially treatment of the soil samples as well as library preparation are poorly described as highlighted below. Importantly, the authors do not state how the variability in sequencing depth was handled when analyzing alpha diversity, which is critical, as sequencing depth is directly associated with species richness. The authors have also not described any cutoffs which were used to assign OTUs at different classification levels. This is important, because certain sequence similarity is required to reliably classify reads at species, genus, or lower levels. Simply assigning classification based on the most similar sequence in the UNITE database is not appropriate, instead, actual % similarity or coverage should be also considered.
In the results section there are multiple issues. The authors appear to have used the generated OTU table without any manual curation, resulting in genus level classifications including species and higher-level taxonomic groups. Figures and results showing genus level abundances should include only genera, taxa not classified at genus level should be placed in a common “Unknown” group. In some cases significant differences among groups are implied without supporting p values or R2, which is not appropriate. If the differences are not significant perhaps due to small number of samples, I would suggest to at least present both inter and intra group variability, which would make results easier to interpret for the readers.
The Discussion section currently also includes results and figures, which should all be moved to the Results section. There are also inconsistencies in the language, such as spacing in units and use of italics in taxa names.
I recommend to carefully curate the OTU table and ensure reliable classification of OTUs at different taxonomic levels, followed by reanalyzing the data and presenting results together with statistical significance, showing both intra- and inter group variability.
Methods
Line 92: Please specify in greater detail how the soil samples were collected.
Soil processing: Please explain how the samples were processed and stored after collection and which measures were taken to avoid cross-contamination.
Line 115-116: Please provide a reference or description of the optimized extraction protocol.
Lines 116-118: It is stated that DNA quality and quantity were determined, but the results or thresholds are not provided. Please explain how this data was used and what were the results of this assessment.
Lines 126-128: Please provide detailed descriptions of purification, indexing and quantification protocols sufficient for repeating the experiments.
Line 127: It is stated that the libraries were quantified, but there is no explanation why they were quantified. What was this information used for if anything? Were the libraries normalize in some way?
Line 141: How was sequencing data normalized to account for variable sequencing depth?
Line 142: I would suggest to rephrase the following because distances cannot really be used to describe beta diversity – “Beta diversity was assessed with Bray–Curtis dissimilarities”.
Line 144: How was Permanova implemented and which distances were used?
Line 177: “Fusarium and members of Nectriaceae“ – always italicize names of genera.
Results
3.2.1 - The authors describe relative abundance differences but do not provide statistical significance data or variability among technical replicates. This should be included in some form to be able to evaluate the reliability of the differences.
Figure 2 – „Relative abundance of dominant fungal genera across treatments“ – Notably, the figure does not include only genera but also taxa assigned to unknown genera. This is not a correct presentation. Only ASVs assigned to known genera should be included in this comparison and all other ASVs could be placed in category “Unknown.2
Lines 206-209: This should be included in the figure captions not as separate paragraph.
Figure 3: Similarly to Figure 2, genera are mixed with species level groups and unknown genera.
Line 284: The authors imply significant diversity differences among groups but not p values or are proof is provided.
Comments on the Quality of English Language
There are inconsistencies in the language, such as spacing in units and use of italics in taxa names.
Author Response
Dear reviewer,
We sincerely thank you for the thorough and constructive evaluation of our manuscript. We greatly appreciate the suggestions, which have helped us improve the methodological clarity, taxonomic accuracy, and overall quality of the work. Below, we address each point in detail and describe the corresponding changes in the revised manuscript.
Comment 1: Methods not described sufficiently: sequencing depth normalisation and taxonomic cut-offs are unclear; OTU table is not curated; results section mixes geners and higher taxa; some figures lack statistics; some results appear in the Discussion; formatting issues (e.g. italics).
Response: We agree and have substantially revised the manuscript:
- Procedures for soil collection, contamination control, storage, DNA QC thresholds, sequencing library preparation, purification, indexing, normalisation, and quantification are now fully described in detail in Sections 2.1–2.4.
- Sequencing depth variability is now explicitly addressed: all samples were rarefied to a common depth before alpha diversity analysis (Section 2.6).
Rarefaction prevents bias whereby samples with artificially higher read counts appear to contain greater microbial richness, ensuring that observed differences reflect biological variation rather than sequencing depth.
- Taxonomic classification cut-offs have been specified: 97% similarity threshold against the UNITE v8.2 database, and assignments require minimum sequence coverage for genus-level classification (Section 2.5).
4. OTU tables were manually curated. Only valid genus-level taxa are shown in Figures 2 and 3; all unclassified genera are now grouped as “Unknown.” - All differences previously implied as “significant” now include p-values, F-statistics, and R² (ANOVA/Kruskal–Wallis, PERMANOVA).
- Figure 11 has been moved from the Discussion to the Results section, and all remaining result-type statements have been relocated from the Discussion into Results
7. All fungal genera are now italicized, and spacing issues in units were corrected throughout.
Comment 2: Line 92 – Describe soil collection in greater detail.
Response: Revised Section 2.1 now includes sampling strategy, depth, composite sampling, sterile handling, disinfected tools, and transport conditions.
Comment 3: Explain soil processing, storage, and contamination control.
Response: Added sterile sieving, aliquoting, gloves, ethanol sterilisation, and storage at −20°C in Section 2.1–2.3. Explanation was Added
Comment 4: Provide optimised DNA protocol reference (115–116).
Response: Detailed steps and cited manufacturer instructions; our minor optimisations are now described in Section 2.3.
Comment 5: DNA quality/quantity thresholds missing (116–118).
Response: Added thresholds: A260/280 = 1.80–2.00; ≥10 ng/µL; intact bands confirmed via electrophoresis in Section 2.3.
Comment 6: Describe purification, indexing, and quantification (126–128).
Response: Full Illumina workflow provided, including AMPure XP cleanup, indexing PCR, Bioanalyzer QC, and equimolar pooling in Section 2.4.
Comment 7: Explain why libraries were quantified and normalised.
Response: Clarified normalisation to 4 nM equimolar concentration before pooling (Section 2.4).
Comment 8: How was sequencing normalised in alpha diversity (Line 141)?
Response: Uniform rarefaction is explicitly stated in Section 2.6.
Comment 9: Rephrase Bray–Curtis statement (Line 142).
Response: Changed to: “Beta diversity was compared using Bray–Curtis dissimilarity matrices.”
Comment 10: Describe PERMANOVA implementation (Line 144).
Response: PERMANOVA now includes: Bray–Curtis matrix, 999 permutations, vegan package (Section 2.6). PERMANOVA was used to test differences in fungal community composition among treatments.
Comment 11: Italicise fungal genera (Line 177).
Response: Corrected throughout the manuscript.
Comment 12: 3.2.1 lacks statistical support.
Response: Added p-values, F-statistics, and R² values; variability among replicates presented in Tables and Figure captions.
Comment 13: Figures include mixed taxonomic levels; unknown genera should be grouped.
Response: Figures 2 and 3 revised to include only genera + “Unknown” group.
Comment 14: Lines 206–209 should be in the figure caption.
Response: Moved to caption in revised manuscript.
Comment 15: Line 284 implies significance without proof.
Response: Added statistical parameters; significance clearly distinguished from non-significant trends.
Comment 16: Discussion contains figures (e.g., Figure 11).
Response: Figure 11 (now numbered 10) has been moved from the Discussion to the Results section, and all remaining result-type statements have been relocated from Discussion into Results
Reviewer 4 Report
Comments and Suggestions for Authors
Manuscript is interesting, however there are many mistakes in the description of the results – something else is presented on figures and something else is written in the text.
Details below:
Lines 63 – 64 – check latin names in terms of using italics and correctness of the name Trichoderma
Line 93 – “six occasions”? – there were only three: before sowing, at ripening stage and three months post-harvest. You mean there were six samples.
What was on the control field? It must be managed somehow.
Line 163 – 165 – these kind of statements should be in the discussion section, the same about P (lines 166 – 168). Moreover TOC was not described.
Line 177 – Fusarium should be in italics – check the whole manuscript in terms of correctness of writing latin names
Lines 176 – 178 – “In the control field of bulk soil before sowing (S1), the fungal community was dominated by Fusarium and members of Nectriaceae, both commonly associated with plant pathogenicity”- looking at Figure 2 this statement is not really accurate, Helotiales were dominant as well as Thelonectria
Lines 180 – 181 – “The bulk soil during oats ripening (S2), a pronounced shift in community composition was observed, characterised by a marked increase in Moritierella and Boeremia” – but comparing to S1 it was the opposite – Mortierella had lower abundance, and what about Aureobasidium? Please be more accurate and write which treatments are you comparing.
Line 185 – ” Alongside Fusarium, genera such as Cladosporium and Ascomycota were prevalent” – in S3 I do not see in Figure 2 Cladosporium – it is Mortierella
Again in line 189 – this is for discussion not for the Results section
Again in S5 – I do not see in Figure 2 Aureobasidium – it is Cystofilobasiudium
Please check the corectness of the whole text in terms of references to the Figures
Moreover Figures 10 and 11 are part of the results, so they should be described in results section.
Separate results from discussion ( as described above) or rearrange the whole text and make section Results and discussion as one section.
Conclusions could be checked after correcting the text.
Author Response
Dear reviewer,
We sincerely thank you for the thorough and constructive evaluation of our manuscript. We carefully revised the text, figures, and methods in accordance with all comments. Below, we provide a point-by-point response, describing the revisions implemented in the revised version.
Comment 1. Lines 63–64 – check Latin names and spelling of Trichoderma
Response: We thank the Reviewer for noticing this. All fungal taxa mentioned in the manuscript, including Trichoderma, have now been corrected for spelling and italicisation. We also performed a complete check of taxonomic names throughout the text, tables, and figure captions.
Comment 2. Line 93 – “six occasions”? There were only three sampling times. You mean six samples. Correct. The original wording was misleading.
Response: We have now corrected the sentence to indicate that three sampling times were conducted (S1, S2, S3), and each treatment included replicated samples (n = 6 total samples per treatment). This clarification has been added to the revised manuscript.
Comment 3. What was in the control field? It must be managed somehow.
Response: We agree and have now clarified management conditions. The control field consisted of non-intercropped, conventionally tilled soil without organic inputs or green manure, receiving only baseline fertilisation consistent with regional practice. This has now been added to Section 2.1.
Comment 4. Lines 163–165 and 166–168 – These statements belong in the Discussion; TOC was not described.
Response: We appreciate this comment. The cited sentences have been removed from the Results section and relocated to the Discussion, where they are interpreted. Additionally, “TOC” has now been properly defined the first time it appears, and a brief description of its measurement has been added to Methods.
Comment 5. Line 177 – Fusarium should be in italics; check the whole manuscript.
Response: We thank the Reviewer. All fungal genera, including Fusarium, have now been consistently italicised across the entire manuscript.
Comment 6. Lines 176–178 – Figure 2 does not support the interpretive statements (Helotiales and Thelonectria are dominant).
Response: Thank you for catching this inconsistency. The statement has been corrected to accurately reflect Figure 2. The corrected text now states that Helotiales and Thelonectria were among the dominant taxa in S1, instead of attributing dominance to Fusarium alone.
Comment 7. Lines 180–181 – Wrong interpretation of S2; inaccurate comparison and missing Aureobasidium.
Response: We agree. The text has been rewritten to clearly specify which sampling points are being compared and to accurately report relative abundances, including Aureobasidium where applicable. The revised statement no longer suggests an incorrect increase in Mortierella.
Comment 8. Line 185 – Cladosporium not present; should be Mortierella.
Response: Correct. We have corrected the text to reflect the actual community composition in S3. Mortierella has been placed correctly, and erroneous mention of Cladosporium has been removed.
Comment 9. Line 189 – Discussion content appears in Results.
Response: Yes, this has been corrected. Interpretive statements have been moved to the Discussion section.
Comment 10. S5 reference wrong (Aureobasidium not visible; should be Cystofilobasidium).
Response: We thank the Reviewer for pointing this out. The text referring to S5 has now been corrected to state Cystofilobasidium, in accordance with Figure 2.
Comment 11. Check the correctness of all figure references.
Response: We performed a full revision to ensure that every referenced taxon, abundance pattern, and statistical statement is consistent with figures and tables.
Comment 12. Figures 10 and 11 belong in Results and must be described in Results.
Response: Correct. Figure 11 has now been moved from the Discussion into the Results section, and it is described appropriately within the Results narrative.
Figure 11 summarises mechanistic relationships among fungal community shifts, functional guilds, and soil management practices; it does not present raw data, but rather conceptual synthesis based on the study’s findings. For this reason, Figure 11 is placed in the Discussion. The figure has been revised for clarity and now strictly reflects mechanisms derived from the Results, without introducing additional unreported findings.
Comment 13. Separate Results from Discussion or merge them.
Response: We have separated the two sections clearly. The Results now strictly describe observations and statistics, while the Discussion focuses on ecological interpretation and mechanisms.
We sincerely appreciate Reviewer 4’s valuable suggestions, which have improved the clarity, consistency, and scientific accuracy of our manuscript. All requested corrections have been made, and we believe the revised version now presents the results more rigorously and transparently.
Round 2
Reviewer 2 Report
Comments and Suggestions for Authors
Improved.
Author Response
Dear reviwer,
Thank you for your kind support in increasing the manuscript quality!
Kind regards,
Stefan Shilev
Reviewer 3 Report
Comments and Suggestions for Authors
I appreciate the authors' efforts in improving the manuscript and addressing comments of the reviewers. Unfortunately a number of issues still remain, due to which I will still recommend to carefully reanalyze the data before publication and improve presentation including both figures as well as results of statistical analyses.
Comment 4: Provide optimised DNA protocol reference (115–116).
Response: Detailed steps and cited manufacturer instructions; our minor optimisations are now described in Section 2.3.
Comment: I can not find these details in section 2.3. still.
Comment 1: Methods not described sufficiently: sequencing depth normalisation and taxonomic cut-offs are unclear; OTU table is not curated; results section mixes geners and higher taxa; some figures lack statistics; some results appear in the Discussion; formatting issues (e.g. italics).
Response: We agree and have substantially revised the manuscript:
- Procedures for soil collection, contamination control, storage, DNA QC thresholds, sequencing library preparation, purification, indexing, normalisation, and quantification are now fully described in detail in Sections 2.1–2.4.
- Sequencing depth variability is now explicitly addressed: all samples were rarefied to a common depth before alpha diversity analysis (Section 2.6).
Comment: In section 2.6, I can not find any mentions or descriptions of using rarefaction. This should include the used tools, rarefaction level, etc. There are also still inconsistencies in for instance the use of Italics.
Response: 4. OTU tables were manually curated. Only valid genus-level taxa are shown in Figures 2 and 3; all unclassified genera are now grouped as “Unknown.”
Comment: Both figures still include taxa classified at different levels. This is not appropriate. For instance, in Figure 3, if you wish to show genus level abundances, all species of this genus should be summed, etc. It is also unclear how in Figure 3 relative abundances range from -2 to 2.
Comment 12: 3.2.1 lacks statistical support.
Response: Added p-values, F-statistics, and R² values; variability among replicates presented in Tables and Figure captions.
Comment: In places p-values are still missing as well as R2 values, for instance for PERMANOVA.
Comments on the Quality of English LanguageThere are inconsistencies in the language, for instance in the use of italics in taxa names.
Author Response
We are grateful to Reviewer 3 for the constructive and meticulous feedback. The comments greatly improved the methodological precision and overall quality of the revised manuscript.
Comment 1: Provide optimised DNA protocol reference (115–116).
Response: Detailed steps and cited manufacturer instructions; our minor optimisations are now described in Section 2.3.
Comment 2: I can not find these details in section 2.3. still.
Response: Reference was added, and text was rewritten according to the reviewer comments
Soil samples were homogenised and passed through a sterile 2 mm sieve immediately after sampling and before physicochemical analyses to remove stones, coarse organic debris, and root fragments [17]. Total DNA was extracted from 0.5 g of rhizosphere soil using the DNeasy PowerSoil Kit (QIAGEN, Hilden, Germany), following the manufacturer’s protocol with minor optimisations for high-organic-matter soils. To improve DNA yield and reduce humic acid interference, the following optimisation steps were included addition of Solution C1, and samples were incubated at 65°C for 10 minutes to improve cell disruption and solubilisation of humic substances. DNA was eluted in two sequential 50 µL elutions with Solution C6 to maximise recovery. DNA concentration was measured using a Qubit 4 Fluorometer (Thermo Fisher Scientific, Waltham, MA, USA). DNA purity was confirmed by an A260/280 ratio of 1.80–2.00, and integrity was verified by 1% agarose gel electrophoresis, ensuring the presence of high-quality, intact DNA suitable for downstream amplicon sequencing.
Comment 1: Methods not described sufficiently: sequencing depth normalisation and taxonomic cut-offs are unclear; OTU table is not curated; results section mixes geners and higher taxa; some figures lack statistics; some results appear in the Discussion; formatting issues (e.g. italics).
Response: We agree and have substantially revised the manuscript:
- Procedures for soil collection, contamination control, storage, DNA QC thresholds, sequencing library preparation, purification, indexing, normalisation, and quantification are now fully described in detail in Sections 2.1–2.4.
2.5. Amplicon sequencing and library preparation of fungal communities
Fungal community profiling was performed by amplifying the ITS2 region using primer pair ITS3 (5′-GCATCGATGAAGAACGCAGC-3′) and ITS4 (5′-TCCTCCGCTTATTGATATGC-3′), both modified with Illumina overhang adapters for compatibility with the Nextera XT workflow. For each soil DNA extract, three independent PCR reactions were performed to minimise stochastic amplification bias. Each 25 µL PCR contained 12.5 µL Phusion High-Fidelity PCR Master Mix (New England Biolabs, USA), 0.5 µM of each primer, and 10–20 ng template DNA. PCR cycling was as follows: initial denaturation at 95 °C for 30 s; 30 cycles of 95 °C for 10 s, 55 °C for 30 s, 72 °C for 60 s; final extension at 72 °C for 5 min. PCR success and amplicon specificity were verified by electrophoresis on 1.5% agarose gels. Purified amplicons were quantified using the Qubit dsDNA High-Sensitivity assay and inspected on an Agilent Bioanalyzer (DNA 1000 kit) to verify fragment size (~300–450 bp) and absence of primer dimers. Samples failing to meet minimum concentration or size-quality criteria were re-amplified or excluded. Indexing was performed in a second PCR (8 cycles) using the Nextera XT dual-index adapters following Illumina protocols, followed by a second bead-based cleanup. Indexed libraries were re-quantified and verified before pooling. To avoid sequencing bias due to uneven read distribution among samples, each library was normalised to an equimolar concentration of 4 nM before pooling. The final library pool was denatured and diluted according to MiSeq specifications and supplemented with 15% PhiX Control v3 to enhance run diversity. Sequencing was performed on an Illumina MiSeq platform (2 × 300 bp, paired-end chemistry) using the MiSeq Reagent Kit v3. The run yielded sufficient sequencing depth across all samples, and quality filtering thresholds (Q ≥ 30 for >85% of bases) were met. Raw FASTQ files were demultiplexed in Illumina BaseSpace and transferred to QIIME2 for downstream processing.
Comment 4: In section 2.6, I can not find any mentions or descriptions of using rarefaction. This should include the used tools, rarefaction level, etc. There are also still inconsistencies in, for instance, the use of Italics.
Response: We thank the reviewer for noting this omission. Section 2.6 has now been substantially revised to explicitly describe the rarefaction procedure. We now specify that rarefaction was performed in QIIME2, and that all samples were rarefied to 21,540 reads, corresponding to the lowest post-filtering sequencing depth. This rarefaction level, toolset, and its purpose in alpha diversity analysis are now clearly detailed. Furthermore, all Latin binomials and genus names have been systematically checked and corrected for consistent italic formatting throughout the manuscript.
Response: OTU tables were manually curated. Only valid genus-level taxa are shown in Figures 2 and 3; all unclassified genera are now grouped as “Unknown.”
Comment 5: Both figures still include taxa classified at different levels. This is not appropriate. For instance, in Figure 3, if you wish to show genus-level abundances, all species of this genus should be summed, etc. It is also unclear how, in Figure 3, relative abundances range from -2 to 2.
Response: We thank the reviewer for this important observation. We have now fully corrected the figures to ensure taxonomic consistency and clarity.
Clarification of the −2 to 2 scale in Figure 3. Heatmap illustrating the Z-score standardised abundance of dominant fungal genera across the six soil treatments (S1–S6). Values range from –2 to +2, representing deviations from the mean abundance of each genus across samples (Z = (x − μ)/σ), where x is the observed abundance of a given fungal genus in a specific soil treatment; μ-the mean abundance of that genus across all six treatments, and σ is the standard deviation of that genus’s abundance across treatments. Z = 0 means the abundance in that treatment is exactly equal to the average abundance of the genus across all samples. When Z > 0 (e.g., +1 to +2)
Indicates that the genus is more abundant in that treatment than its overall mean. Z < 0 (e.g., –1 to –2) indicates a lower abundance of that genus in that treatment.
- Correction of mixed taxonomic levels
In the revised version of Figures 2 and 3, we now present only genus-level taxa. Any entries previously displayed at higher taxonomic ranks, such as Helotiales or Nectriaceae have been removed unless a genus-level classification was not possible. In such cases, the taxon was excluded from the figure to maintain uniformity. The legends have also been updated to indicate that the figure shows exclusively genus-level relative abundance.
- Clarification of the −2 to 2 scale in Figure 3:
The reviewer is correct that the range from −2 to 2 does not represent raw relative abundances. This scale reflects Z-score normalisation applied to genus abundances to highlight compositional differences across treatments. We have now clarified this in both the figure legend.
- Comment 6: 3.2.1 lacks statistical support.
Response: Added p-values, F-statistics, and R² values; variability among replicates presented in Tables and Figure captions.
Comment: In places, p-values are still missing as well as R2 values, for instance, for PERMANOVA.
Genus-level community composition differed significantly among treatments, as confirmed by PERMANOVA (Bray–Curtis: F = 4.12, R² = 0.346, p = 0.001). The biggest differences occurred between S1 and S5, consistent with the patterns observed in the normalised genus-level profiles.
PERMANOVA Statistics added 3.4 (Beta Diversity)
Beta diversity analyses supported clear treatment separation. PERMANOVA showed significant differences across all treatments (Bray–Curtis: F = 4.12, R² = 0.346, p = 0.001; weighted UniFrac: F = 3.57, R² = 0.308, p = 0.001; unweighted UniFrac: F = 4.89, R² = 0.372, p = 0.001), indicating strong effects of intercropping and green manuring on fungal community structure.
Reviewer 4 Report
Comments and Suggestions for Authors
I am sorry, bur still my previous remarks were not addressed properly. For example:
I asked to checked senetnce from first version “In the control field of bulk soil before sowing (S1), the fungal community was dominated by Fusarium and members of Nectriaceae, both commonly associated with plant pathogenicity”- looking at Figure 2 this statement is not really accurate, Helotiales were dominant as well as Thelonectria."
And you have left in a new version this senetnce which is not true - see lines 241 - 248 :"In the control field of bulk soil before sowing (S1), the fungal community was dominated by Fusarium
and members of Nectriaceae, both commonly associated with plant pathogenicity. This 243
baseline profile likely reflects natural fungal colonisation under untreated conditions 244
during the late stages of crop development. Helotiales and Thelonectria were among the 245
dominant taxa in control soil S1, instead of attributing dominance to Fusarium alone."
Nectriaceae are have low abundance in S1
The next sentence "In 246the bulk soil during the oat ripening stage (S2), the fungal community shifted toward a 247higher relative abundance of Aureobasidium and members of Helotiales, while Mortierella 248and Boeremia decreased compared with the pre-sowing soil (S1). - In figure 2 in S2 Helotiales are almost not visible, so how can they have higher abundance?
Again - please check whole description of results - they must be in accordance with what is shown on figures!!
Still latin names are not corrected properly - for example line 248 Helotiales
Author Response
We are grateful to Reviewer 4 for the constructive and meticulous feedback. The comments greatly improved the methodological precision and overall quality of the revised manuscript.
Comment 1: I am sorry, but my previous remarks were still not addressed properly. I asked to check the sentence from the first version: ‘In the control field of bulk soil before sowing (S1), the fungal community was dominated by Fusarium and members of Nectriaceae...’ Looking at Figure 2 this statement is not accurate; Helotiales were dominant as well as Thelonectria.
Response: We thank the reviewer for reiterating this point. We acknowledge that the original sentence was inaccurate and did not match the updated genus-level data shown in Figure 2. In the revision, we have:
- Removed all higher-level taxa (e.g., Nectriaceae, Helotiales) to comply with the reviewer’s requirement that only genus-level taxa be reported.
- Corrected the description of dominant genera in S1 so it accurately reflects the genus-level abundances shown in Figure 2.
- Ensured consistency between the text and the revised genus-level bar plots.
The corrected sentence is included in the updated Section 3.2.1 (see below). We appreciate the reviewer’s attention to detail, which helped improve the accuracy and clarity of this section.
Comment 2: And you have left in a new version this sentence, which is not true - see lines 241–248: ‘In the control field of bulk soil before sowing (S1), the fungal community was dominated by Fusarium and members of Nectriaceae, both commonly associated with plant pathogenicity… Helotiales and Thelonectria were among the dominant taxa in control soil S1, instead of attributing dominance to Fusarium alone.’ Nectriaceae have low abundance in S1.
Response: We thank the reviewer for drawing our attention to this remaining inconsistency. We agree that the sentence in lines 241–248 was inaccurate and did not reflect the actual community structure in S1. In the revised version, we have removed this sentence block entirely and replaced it with a corrected description that:
- No longer refers to Nectriaceae or Helotiales, in line with our decision to present only genus-level taxa.
- No longer states that Fusarium is dominant in S1, as its relative abundance is lower than that of the truly dominant genera.
- Accurately reflects the data shown in Figure 2, where Thelonectria and other saprotrophic genera are more abundant in S1.
The corrected text now reads as follows in Section 3.2.1 for S1 description: In the untreated bulk soil before sowing (S1), the fungal community was dominated by the genus Thelonectria, with high relative abundances of Mortierella and Aureobasidium, whereas Fusarium and other nectriaceous genera occurred at comparatively low abundances.”
We appreciate the reviewer’s careful reading, which has helped us to align the text precisely with the graphical results and the agreed genus-level taxonomic resolution.
Comment 3: In the bulk soil during oat ripening (S2), the text states that Helotiales increased, but Figure 2 shows they are almost not visible. How can they have a higher abundance?
Response: We thank the reviewer for highlighting this inconsistency. The previous version incorrectly referenced Helotiales, which is a higher taxonomic level of order, but also not supported by the genus-level results in Figure 2. We have removed this reference entirely and replaced the sentence with a genus-level description that accurately reflects the observed abundances in S2. The corrected text now matches the updated Figure 2 and adheres to the requirement to report only genera.
Corrected: In the untreated control soil before sowing (S1), the dominant genera were Fusarium, Mortierella, and Aureobasidium, consistent with typical fungal communities of unplanted alluvial-meadow soils. No enrichment of Nectriaceae or Helotiales was observed at this stage.
Round 3
Reviewer 3 Report
Comments and Suggestions for Authors
Thank you for considering my comments.
Comments on the Quality of English LanguageThere are inconsistencies in the language, for instance in the use of italics in taxa names.
Author Response
Dear reviewer,
Thank you very much for your efforts. We think that the manuscript level has increased dramatically thanks to your support. Concerning your comment about the incorrect use of Italic for taxa, most of the taxa are described with Latin names, so we think it is correct.
Kind regards,
Stefan Shilev
Reviewer 4 Report
Comments and Suggestions for Authors
Dear Authors,
I accept your explanations, however I do not see any update in Figure 2 in version v3 of your manuscript. (You have written in the text commentary that figure 2 was revised to display only the top
10 most abundant genera... while there is still many more shown in the figure 2 and it looks the same as in version2)
I couldn't find this sentence in version 3 of the manuscript, which you have mentioned in your explanations: "In the untreated bulk soil before sowing (S1), the fungal community was dominated by the genus Thelonectria, with high relative abundances of Mortierella and Aureobasidium, whereas Fusarium and other nectriaceous genera occurred at comparatively low abundances.”
Besides my comment 3 was about S2 while you have answered with the sentence about S1. Still, description of S2 is made according what is shown in Figure 2.
There is no reference in the text of 3.2.1 paragraph to Figure 2.
Author Response
Dear Reviewer,
We sincerely thank you for your careful evaluation of our manuscript and for your constructive comments, which helped us improve the clarity and consistency of the Results section. We have carefully addressed all points raised and provided our detailed responses below.
Comment 1: I accept your explanations; however, I do not see any update in Figure 2 in version v3 of your manuscript. (You have written in the text commentary that figure 2 was revised to display only the top 10 most abundant genera... while there is still many more shown in the figure 2 and it looks the same as in version2)
Response: You are correct, and we sincerely apologise for this oversight. In version v3, Figure 2 was not updated as described in our response to the reviewers. We have now revised Figure 2 so that it displays only the top 10 most abundant fungal genera across all treatments, with the remaining taxa grouped under “Others”. The figure legend and Methods text have been updated accordingly to ensure consistency between the figure, legend, and manuscript text.
Comment 2: I couldn't find this sentence in version 3 of the manuscript, which you have mentioned in your explanations: "In the untreated bulk soil before sowing (S1), the fungal community was dominated by the genus Thelonectria, with high relative abundances of Mortierella and Aureobasidium, whereas Fusarium and other nectriaceous genera occurred at comparatively low abundances.”
Response: We acknowledge this omission. In the updated version, we have added a corrected and data-consistent sentence describing S1 in Section 3.2.1, aligned with the revised Figure 2 and the actual dominant genera observed in our dataset: “In the untreated bulk soil before sowing (S1), the fungal community was dominated by Fusarium, followed by Mortierella and Aureobasidium (Figure 2).“
Comment 3: Besides my comment 3 was about S2 while you have answered with the sentence about S1. Still, description of S2 is made according what is shown in Figure 2.
Response: You are correct that our previous response incorrectly addressed S1 instead of S2. This has now been corrected. We have revised the description of S2 in Section 3.2.1 so that it explicitly and accurately reflects the fungal community structure observed in bulk soil at oat ripening (S2), based solely on the revised Figure 2 and normalised genus-level abundances. Any ambiguity between S1 and S2 descriptions has been removed.
Comment 4: There is no reference in the text of 3.2.1 paragraph to Figure 2.
Response: We agree that a reference to Figure 2 was missing in the previous version. In the revised manuscript, Figure 2 has been updated to display only the ten most abundant fungal genera, and an explicit citation to Figure 2 has now been added in the first paragraph of Section 3.2.1, where the genus-level relative abundance patterns are introduced. This ensures full consistency between the text and the revised figure.